# BENCHHUB: A UNIFIED BENCHMARK SUITE FOR HOLISTIC AND CUSTOMIZABLE LLM EVALUATION

## ABSTRACT

As large language models (LLMs) continue to advance, the need for up-to-date and well-organized benchmarks becomes increasingly critical. However, many existing datasets are scattered, difficult to manage, and make it challenging to perform evaluations tailored to specific needs or domains, despite the growing importance of domain-specific models in areas such as math or code. In this paper, we introduce BENCHHUB, a dynamic benchmark repository that empowers researchers and developers to evaluate LLMs effectively, with a focus on Korean and English. BENCHHUB aggregates and automatically classifies benchmark datasets from diverse domains, integrating 839k questions across 54 benchmarks. It is designed to support continuous updates and scalable data management, enabling flexible and customizable evaluation tailored to various domains or use cases. Through extensive experiments with various LLM families, we demonstrate that model performance varies significantly across domain-specific subsets, emphasizing the importance of domain-aware benchmarking. Furthermore, we extend BENCHHUB into 10 languages spanning resource levels. We believe BenchHub can encourage better dataset reuse, more transparent model comparisons, and easier identification of underrepresented areas in existing benchmarks, offering a critical infrastructure for advancing LLM evaluation research.

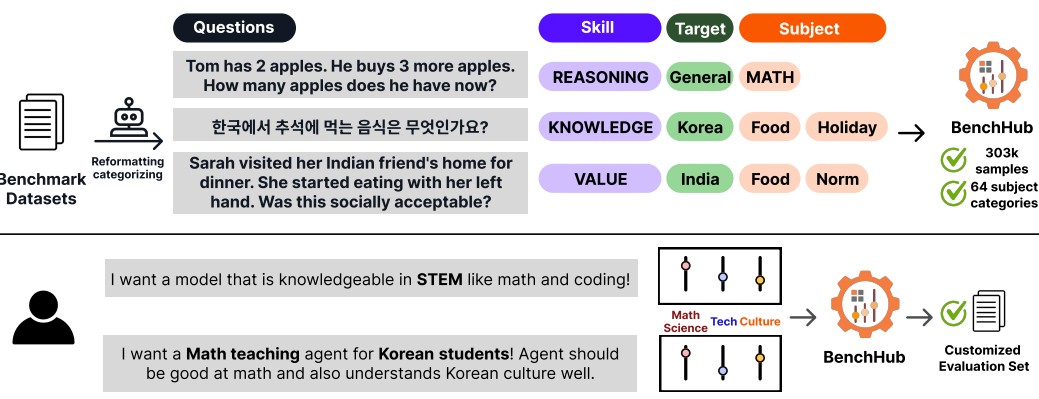

Figure 1: The concept of BENCHHUB. BENCHHUB automatically classifies and merges questions from existing benchmark datasets on a sample-wise basis. Through BENCHHUB, users can select test sets that align with their objectives and efficiently evaluate the models.

## 1 INTRODUCTION

Large language models (LLMs) have made remarkable strides, powering applications across diverse tasks, including research (Baek et al., 2025), industry (Chan et al., 2025), and everyday life (Chatterji et al., 2025). As their role varies with context—expanding into open-ended and high-stakes challenges ranging from utilizing external tools (Yang et al., 2023a) to making culturally sensitive decisions (Ki et al., 2025)—a new paradigm for LLM evaluation is essential. The key question is moving beyond

formulaic rankings toward a rigorous and comprehensive assessment of whether a model's behavior aligns with the nuanced, custom objectives of specific users and applications.

In response, a wide range of evaluation efforts has emerged. On the one hand, holistic evaluation benchmarks (Liang et al., 2023; Ni et al., 2024) and leaderboards based on user preference (Chiang et al., 2024) or aggregated benchmarks (Aidar Myrzakhan, 2024) serve as popular community standards. While useful for broad comparisons, their aggregated scores obscure fine-grained strengths and weaknesses, often misaligning with the needs of specific applications Ribeiro et al. (2020). On the other hand, specialized benchmarks target narrow aspects, such as law (Li et al., 2024a), medical advice (Arora et al., 2025), and finance (Son et al., 2023), as well as specific tasks, including knowledge retrieval (Hendrycks et al., 2021a), reasoning (Cobbe et al., 2021; Zellers et al., 2019), and value alignment (Parrish et al., 2022; Ji et al., 2024). While these datasets capture critical capabilities, their vast, fragmented, and overlapping nature creates a chaotic landscape. For instance, in the mathematics domain, numerous benchmarks exist, such as MATH (Hendrycks et al., 2021b) and GSM8k (Cobbe et al., 2021), which in turn partially overlap with broader collections (*e.g.*, MMLU (Hendrycks et al., 2021a)). This leaves researchers and practitioners with a dilemma: which benchmarks truly reflect their objective, and how can they compose a principled, customized evaluation suite tailored for diverse needs?

In this paper, we introduce BENCHHUB [1], a unified and customizable benchmark suite for holistic yet domain-aware LLM evaluation. BENCHHUB aggregates 839k questions from 54 benchmarks across 64 domains and 10 languages, mainly in English and Korean. We systematically categorize existing benchmarks by six dimensions: 1) tasks (*e.g.*, mathematical reasoning), 2) answer formats (*e.g.*, multiple-choice QA), 3) tool usage (*i.e.*, language-only or requirements to external tools), 4) skills (*i.e.*, knowledge, reasoning, or value/alignment), 5) coarse- and fine-grained subjects (*e.g.*, STEM–mathematics), and 6) cultural-specificity (*i.e.*, culturally specific or agnostic). This design facilitates users to dynamically construct their own evaluation sets tailored to their needs, moving beyond rigid, predefined test sets (Figure 1). To ensure long-term, dynamic scalability, we further train and release a categorization model that seamlessly integrates new, unseen benchmarks into BENCHHUB.

Using BENCHHUB, we evaluate 14 open LLMs and uncover a crucial insight: model rankings fluctuate substantially depending on benchmark compositions and domain focus. This finding highlights the central issue of benchmark composition bias, which can significantly distort interpretations of model performance. We further validate BENCHHUB through 5 real-world use cases—such as legal, educational, and cultural applications—showing how domain-aware evaluation alters conclusions about model superiority. We hope BENCHHUB provides a foundation for the community to move beyond monolithic leaderboards toward domain-aware, trustworthy, and customizable evaluation.

## 2 EXISTING LLM EVALUATION BENCHMARKS ARE SKEWED

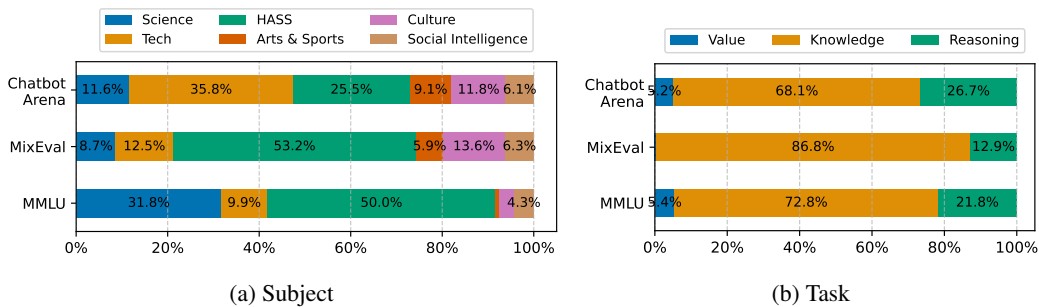

(a) Subject          (b) Task

Figure 2: Data distribution of existing evaluation benchmarks.

---

[1] We release our code and datasets at `https://anonymous.4open.science/r/BenchHub_review-0A86`; due to repository size limitations, only a subset is included, and the full dataset will be made available on Hugging Face following the anonymous review period.

What aspects do the commonly used multi-domain datasets evaluate, and how is the distribution of domains represented across these datasets? To answer this question, we classify three representative holistic benchmarks (*i.e.*, Chatbot Arena (Chiang et al., 2024), MixEval (Ni et al., 2024), and MMLU (Hendrycks et al., 2021a)) as multilabels using our fine-tuned classifiers (§ 3) in terms of coarse-grained subjects (Figure 2a) and tasks (Figure 2b). Among them, Chatbot Arena includes only 25.5% of Humanities and Social Science (HASS) questions, while both MixEval and MMLU comprise more than half of HASS questions. In addition, MixEval includes fewer than 0.30% of value alignment tasks and mostly focuses on measuring knowledge. Such disparities may lead to biased findings, where models that excel in certain domains may appear to perform better overall, potentially skewing the evaluation results.

Moreover, these biases are not limited to cross-benchmark comparisons but can also manifest within multilingual contexts. Figure 3 and Figure 12 illustrate data distributions of MMLU series datasets in 5 languages classified by the model (§ 3) in terms of coarse-grained subjects. For instance, MMLU in English emphasizes HASS, whereas Korean MMLU (KMMLU) (Son et al., 2025b) comprises 76.1% of STEM (Science, Technology, Engineering, and Mathematics) questions. This variation complicates the interpretation of performance differences, as it is challenging to discern whether the performance degradations in non-English are due to language proficiency or domain-specific knowledge.

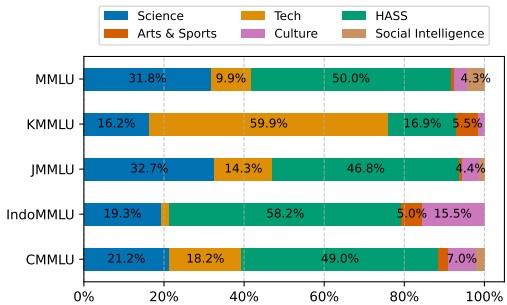

Figure 3: Data distribution of MMLU series in English, Korean, Japanese, Indonesian, and Chinese, respectively

Hence, instead of recklessly adopting existing holistic benchmarks, we recommend carefully selecting the benchmark suites for a reliable evaluation.

## 3 BENCHHUB

Consider a user who wants to determine "*Which model excels at both mathematics and understanding culture?*" As discussed in § 2, it remains unclear how to answer such specific, goal-oriented questions and how to construct their evaluation suite, as existing evaluation benchmarks (Hendrycks et al., 2021a; Liang et al., 2023; Ni et al., 2024) mainly provide general-purpose scores. To this end, we introduce BENCHHUB, a unified collection of LLM evaluation benchmarks across diverse domains. BENCHHUB integrates 54 benchmarks comprising 839k samples in 10 languages, with a primary focus on English and Korean as BENCHHUB-En and BENCHHUB-Ko, respectively. We design BENCHHUB around two core principles: 1) a fine-grained, multi-dimensional taxonomy to deconstruct model capabilities and 2) a fully automated pipeline to dynamically update and expand it with new datasets. In this section, we detail the taxonomy design (§ 3.1), the data curation (§ 3.2), the automated pipeline (§ 3.3), as well as interactive tools and utilities as a web-based platform(§ 3.4). Finally, we illustrate the multilingual extension of BENCHHUB —from English and Korean to eight additional languages—in § 3.5.

### 3.1 TAXONOMY

We annotate each dataset with six orthogonal dimensions: three dataset-level attributes—**task, answer format, and tool usage**— and three sample-level attributes—**skill, subject, and cultural-specificity**. The full scheme is illustrated in Appendix D.

**Dataset-level attributes:**

1. **Task** refers to the high-level family defined by the dataset authors (*e.g.*, mathematical reasoning, code generation, cultural understanding). This provides a general understanding of a dataset's purpose. We assign it automatically from the dataset's abstract or description using LLM inference.

2. **Answer format** specifies the expected response format: binary, multiple-choice QA (MCQA), short-form, free-form, open-ended (*e.g.*, story generation), and comparison (*e.g.*, determining

which response is better between A and B). This is crucial for selecting appropriate evaluation prompts and formats.

3. **Tool Usage** indicates whether a task requires language capabilities only (*language-only*) or interaction with external tools such as *e.g.*, code interpreters, web browsers, calculators (*requires externals tools*). This dimension supports agentic evaluation, where models must decide when and how to invoke external resources.

**Sample-level attributes:**

4. **Skill** captures the required ability to answer the question (*i.e.*, reasoning, knowledge, and value/alignment).

5. **Subject** denotes the knowledge domain. We define six coarse-grained categories—*Science*, *Technology*, *Humanities and Social Science (HASS)*, *Arts & Sports*, *Culture*, and *Social Intelligence*— along with 64 sub-categories, by integrating various knowledge classification systems. Each sample may have multiple subject labels.

6. **Cultural-specificity** represents the cultural or geographical focus. Culturally agnostic items are labeled as *General*; otherwise, we assign a *Local* tag. This supports evaluation under culturally-aware evaluation (Singh et al., 2024).

## 3.2 DATASETS

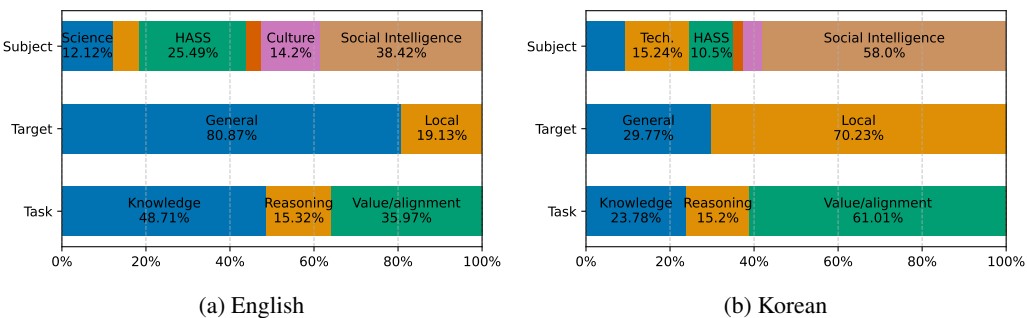

(a) English                (b) Korean

Figure 4: Data distribution of all datasets used in this paper by coarse-grained subjects, targets, and tasks. The English and Korean data include 250,940 and 144,331 questions each.

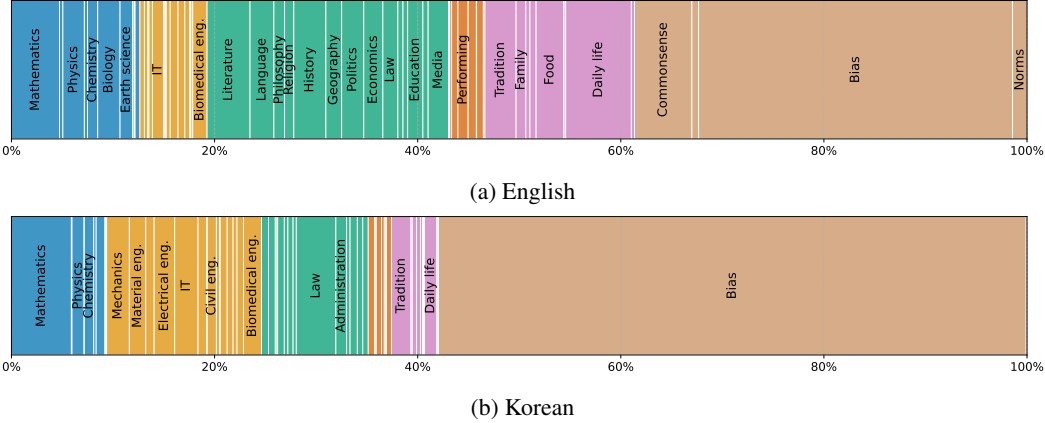

(a) English

(b) Korean

Figure 5: Fine-grained data distribution of all datasets used in this paper in terms of subjects

We apply this taxonomy to 54 benchmarks in 10 languages, mainly covering English and Korean and totaling over 839k instances. Figures 2 and 5 show the overall statistics of English and Korean datasets included in our benchmark. For English and Korean, we include 31 English and 12 Korean

language benchmarks with a total of 41 datasets. [2] We curate 1) general-purpose (*i.e.*, culturally agnostic) datasets commonly used by holistic evaluation benchmarks (Ye et al., 2024; Ni et al., 2024) and 2) culture-specific datasets. We select English datasets spanning multiple cultures drawn from a recent survey (Pawar et al., 2024), curating over 300 papers and datasets regarding LLM cultural awareness. For Korean, where public resources are fewer than in English, we include most datasets released after 2022. Table 2 in the Appendix provides a complete list of the datasets.

### 3.3 AUTOMATED AND DYNAMIC EXPANSION

With benchmark datasets emerging at a rapid pace, it is crucial to flexibly manage them for holistic evaluation. To dynamically adapt to newly emerging datasets, we automate the entire dataset merging process using an LLM agent, which includes reformatting the datasets into our benchmark format and classifying each sample into categories. The processing pipeline for a newly introduced dataset is outlined as follows:

1. **Reformatting:** We first automatically parse, reformat, and map a new dataset to the standardized BENCHHUB scheme using an LLM-guided rule-based approach. If the dataset does not adhere to our predefined schema, an LLM agent (*e.g.*, GPT-4o or Gemini) is employed to map keys to the correct format.

2. **Metadata assignment:** The LLM agent extracts the meta-task description and infers the task, answer format, and tool usage from the dataset documentation (*e.g.*, abstract).

3. **Sample-level Categorization:** We then assign sample-level attributes (*i.e.*, skill, subject, and cultural-specificity) using a fine-tuned Qwen-2.5-7B model (`BenchHub-Cat-7B`).[3]

4. **Merging:** The processed and annotated dataset is seamlessly merged into the main collections, thereby producing the next BENCHHUB release.

This automated pipeline allows BENCHHUB to continuously expand and provide more comprehensive evaluations as new datasets emerge. While we acknowledge the incompleteness of LLM-based expansion, we provide an empirical discussion of the reliability and robustness of this automated process in Appendix E.2.

Tables 1 show the accuracies of the categorizer model in Sample-level Categorization.

Table 1: Accuracy of fine-tuned categorizer on Qwen-2.5-7b

| Sample-level Attribute | Accuracy |
|---|---|
| Subject | 0.871 |
| Skill | 0.967 |
| Cultural-specificity | 0.986 |

### 3.4 INTERACTIVE PLATFORM AND UTILITIES

To proliferate our structured data into actionable insights for researchers and practitioners, we release an interactive web-based platform (Figure 10) and code utilities. The web demo allows users to filter out datasets by any category combinations, inspect statistics, download their customized subsets, and propose new datasets via pull requests. The code utilities offer two main features:

1. **Dataset Loader:** It filters the dataset to include only the categories selected by the user. It also allows the user to choose between returning the entire selected dataset or a filtered version with overlapping entries (including near-duplicates) removed, which is useful since multiple aggregated datasets may contain overlapping samples.

2. **Citation Report Generator:** For the customized dataset returned to the user, it produces a laTeX table of datasets with their sources and licenses, includes dataset statistics such as the number of instances, and provides a comprehensive citation list (e.g., BibTeX entries) to ensure proper credit to dataset authors.

---

[2]We count the multilingual datasets—BLEnD (Myung et al., 2024) and CaLMQA (Arora et al., 2024)—in both.

[3]The model link will be added after the anonymous review period. Details on the training and validation of `BenchHub-Cat-7B` are provided in Appendix E.1.

For better reproducibility, we adopt HRET (Lee et al., 2025) [4], enabling direct evaluations on BENCH-HUB. Design and implementation details of the platform and code utilities appear in Appendix B.

## 3.5 MULTILINGUAL EXTENSION OF BENCHHUB

While we focus on two languages (*i.e.*, Korean and English), we highlight that BENCHHUB is a language-agnostic, flexible framework that can be easily extended to other languages. To empirically guide this extension, we present `BenchHub-Multi-Cat-7B`[5], a multilingual categorizer supporting 10 languages—English (En); 3 high-resource (Arabic (Ar), German (De), Dutch (Nl)); 3 mid-resource (Indonesian (Id), Korean (Ko), Ukrainian (Uk)); 3 low-resource (Swahili (Sw), Nepali (Ne), Kyrgyz (Ky)). Our multilingual categorizer achieves an average accuracy of 77.5% on fine-grained subject categorizations for unseen, out-of-domain data. Furthermore, we introduce BENCHHUB-multilingual, which extends our benchmark suite to a total of 10 languages consisting of 13 datasets and 444,402 samples. We hope BENCHHUB-multilingual to serve as a foundational step for reliable LLM evaluations in non-English languages. The details of the training procedure and the datasets for each language are provided in the Appendix F.

## 4 HOLISTIC AND CUSTOMIZABLE EVALUATION USING BENCHHUB

### 4.1 DOMAIN-AWARE EVALUATION USING BENCHHUB

In this section, we empirically show why the categories provided by BENCHHUB are important for LLM evaluation: performance varies substantially across categories (§ 4.1.1). Consequently, the dataset's category distribution strongly influences model scores and leaderboard rankings (§ 4.1.2).

### 4.1.1 IMPACT OF SUBJECT CATEGORY ON MODEL RANKINGS

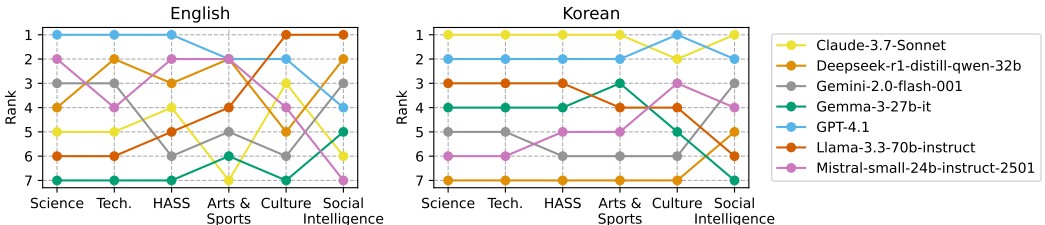

Figure 6: LLM evaluation ranking under BENCHHUB in terms of coarse-grained subjects

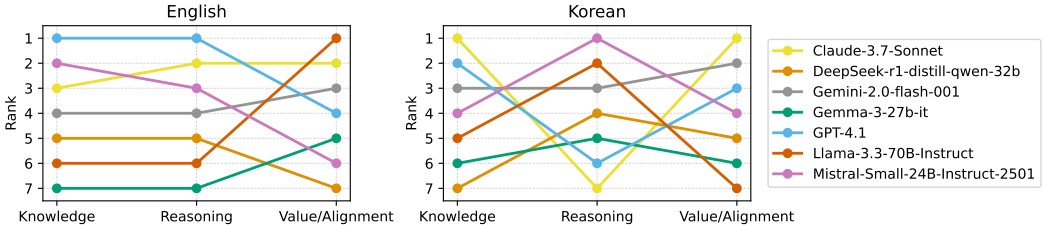

Figure 7: LLM evaluation ranking under BENCHHUB in terms of skills

In this section, we evaluate seven LLMs across diverse subjects using BENCHHUB. We select 6,644 and 6,485 examples for English and Korean, respectively. To manage the large number of fine-grained categories, we sample up to 150 examples per category, fully including categories with 100–150 samples and merging categories with fewer than 80 samples into a miscellaneous group within the same coarse-grained category. We extract the model's intended answer from MCQA questions by

---

[4]HRET is an evaluation toolkit supporting multiple datasets, including BENCHHUB.

[5]The model link will be added after the anonymous review period.

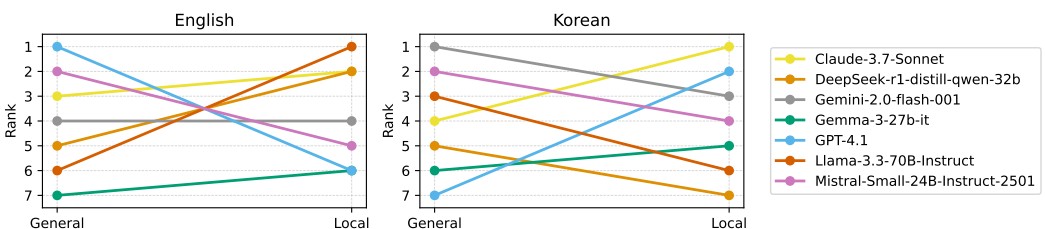

Figure 8: LLM evaluation ranking under BENCHHUB in terms of cultural-specificity

applying a set of regular expressions (Molfese et al., 2025), while using an LLM as a parser extractor for short-form questions [6], similar to the approach in previous work (Ni et al., 2024).

We include one representative model from each commonly used LLM family. For proprietary models, we use GPT-4.1, Gemini-2.0-flash, and Claude 3.7 Sonnet [7]. Open models include Qwen-3-32b (Yang et al., 2025), DeepSeek-R1-Distill-Qwen-32B (DeepSeek-AI et al., 2025), Llama-3.3-70B (Grattafiori et al., 2024), Mistral-Small-24B-Instruct, and gemma-2-27b-it (Team et al., 2025).

Figures 6, 7, 8 present model rankings by subject, skill, and cultural-specificty categories, respectively. Our results show that the rankings fluctuate frequently, depending on the sample-level category. For example, Llama-3.3-70b ranks 6th in Science and Tech., but ranks as the top-performing model in Culture and Social Intelligence. This highlights the importance of domain-aware evaluation aligned with the evaluation context and objectives. The full results on the scores for each subject and model are in Table 17- 18 in the Appendix I.

### 4.1.2 IMPACT OF SAMPLING STRATEGIES ON MODEL RANKINGS

In this section, we empirically validate the influence of category distributions within evaluation benchmarks on model rankings. Since this requires experiments on large datasets for statistical validation, we include 14 open models ranging from 1B to 72B parameters. We test on a diverse set of English and Korean datasets, comprising 16,898 and 18,977 MCQA samples, respectively. The number of answer choices per MCQA sample varies between 3 and 18. We extract the model's intended answer by applying a set of regular expressions (Molfese et al., 2025). The evaluated LLMs include:

- Qwen (Yang et al., 2024; 2025): `Qwen2.5-72B-Instruct`, `Qwen3-1.7B`, `Qwen3-4B`, `Qwen3-8B`, `Qwen3-14B`, `Qwen3-32B`
- DeepSeek (DeepSeek-AI et al., 2025): `DeepSeek-R1-Distill-Qwen-14B`, `DeepSeek-R1-Distill-Qwen-32B`
- Llama (Grattafiori et al., 2024): `Llama-3.1-8B-Instruct`, `Llama-3.3-70B-Instruct`
- Mistral: `Mistral-Small-24B-Instruct-2501`
- Gemma (Team et al., 2025): `gemma-3-1b-it`, `gemma-3-4b-it`, `gemma-3-27b-it`

To gauge the impact of data composition, we experiment under three sampling strategies with four setups, which are representatives of traditional approaches or emerging trends in LLM evaluations with a massive benchmark scale.

- **Random sampling:** Samples are drawn uniformly at random from the entire dataset collection, disregarding category proportions. Each sample has an equal chance of selection.

- **Stratified sampling:** Samples are drawn to ensure equal representation from each constituent dataset, preserving dataset-level balance rather than the overall distribution.

- **Sampling according to category distribution:** This strategy performs stratified sampling guided by fine-grained category distributions observed in existing holistic LLM benchmarks.

---

[6]We use GPT-4.1-nano as a parser extractor. Note that Ni et al. (2024) use GPT-3.5. The LLM parses and compares the extracted answer with the ground truth, without assessing answer quality.

[7]For GPT-4.1, we use GPT-4.1-2025-04-14 version. We directly call GPT-4.1 via the OpenAI API, while we use OpenRouter for Gemini-2.0-flash, and Claude 3.7 Sonnet.

We adopt the distributions derived from Chatbot Arena and MixEval, classified by our fine-tuned model (§ 3.3). The coarse-grained category distributions of these benchmarks are detailed in § 2.

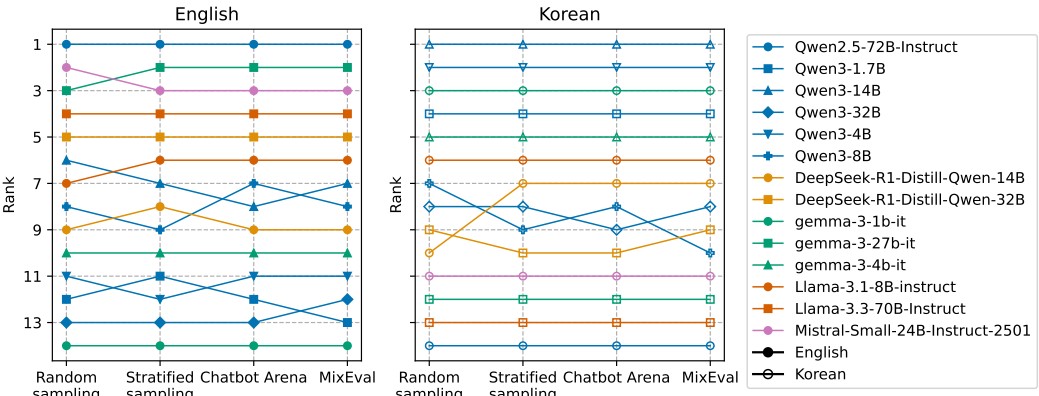

Figure 9: LLM ranking according to four sampling methods

We run 50 simulations per sampling setup, each selecting 5K questions. Model rankings within each setup follow normal distributions. Figure 9 visualizes LLM ranking changes across the four sampling setups. We use the Friedman test and the pairwise Wilcoxon test to statistically identify whether the sampling strategy affects the model ranking based on average accuracy. We observe a statistically significant difference across sampling strategies using the Friedman test ($p < 0.01$). Specifically, pairwise Wilcoxon signed-rank tests confirm that all pairs of sampling setups significantly differ in average, except for random sampling versus sampling according to MixEval distribution ($p < 0.01$). These findings underscore that category distribution and sampling strategy of data substantially affect LLM leaderboard rankings. We call on researchers and practitioners to carefully consider benchmark composition when evaluating LLMs. The full results for each subject and model are in Tables 19 and 20 in the Appendix I.

### 4.2 Customized Evaluation using BenchHub

In this section, we showcase how customized benchmark composition using BenchHub enables more targeted and meaningful evaluations tailored to real-world application scenarios. Here, we consider two use cases illustrated in Figure 1, and construct corresponding customized BenchHub as follows:

(a) **STEM knowledge evaluation:** To identify the best-performing model with expertise in STEM domains, we select English datasets within BenchHub whose coarse-grained subjects are labeled as *Science* or *Technology*. To ensure balanced representation across individual datasets, the questions are drawn using a stratified sampling strategy at a dataset level.

(b) **Math teaching agent for Korean students:** To evaluate Math teaching agents, we select Korean datasets comprising 1) math-related samples (*i.e.*, fine-grained categories are *Science/Math* or *Science/Statistics*), 2) education-related samples (*i.e.*, fine-grained category is *HASS/Education*), and 3) samples culturally specific to Korea (*i.e.*, cultural-specificity as 'KR'). The final accuracy is computed as a weighted average of these subsets, with weights of 0.6, 0.1, and 0.3, respectively, reflecting their relative importance to the application.

Table 2 presents the rankings of LLMs under these customized benchmarks. We use the same set of models described in § 4.1.2. Notably, the model rankings differ substantially depending on the benchmark compositions, underscoring the practical need for tailored evaluations. The full results for each subject and model are in Table 21 in the Appendix I. We provide three additional real-world use cases (*i.e.*, legal chatbot, docent for Korean art, and counseling agent) and their corresponding model results in Appendix G.2.

Table 2: Top-5 LLMs evaluated by BENCHHUB in real-world application scenarios

| Rank | (a) STEM knowledge evaluation (EN) | | (b) Math teaching agent for Korean students (KO) | |
|---|---|---|---|---|
| | Customized | Stratified | Customized | Stratified |
| 1 | Qwen3-32B | gemma-3-1b-it | Qwen2.5-72B-Instruct | Qwen2.5-72B-Instruct |
| 2 | gemma-3-1b-it | Qwen3-32B | Mistral-Small-24B-Instruct-2501 | Llama-3.3-70B-Instruct |
| 3 | Qwen3-1.7B | Qwen3-4B | gemma-3-27b-it | gemma-3-27b |
| 4 | Qwen3-4B | Qwen3-1.7B | Llama-3.3-70B-Instruct | Mistral-Small-24B-Instruct-2501 |
| 5 | DeepSeek-R1-Distill-Qwen-14B | gemma-3-4b | DeepSeek-R1-Distill-Qwen-32B | DeepSeek-R1-Distill-Qwen-32B |

## 5 RELATED WORK

As LLMs have become integral to real-world generative AI systems, the historical focus on benchmarks and leaderboards has matured into evaluation *science* (Weidinger et al., 2025). While LLM evaluation benchmarks primarily adopt a question-answering task as a default evaluation format, they have expanded their capabilities into diverse tasks, including long-form generation (Min et al., 2023), multilingual (Singh et al., 2024; Shafayat et al., 2024), multimodal (Fu et al., 2024), and complex reasoning tasks (Cobbe et al., 2021; Zellers et al., 2019), *inter alia*. This diversification reflects a growing recognition of the multifaceted capabilities and applications of LLMs.

**Domain-specific Evaluation.** Beyond general-purpose benchmarks, there has been a surge in domain-specific evaluation benchmarks targeting verticals such as healthcare and medicine (Hertzberg and Lokrantz, 2024; Matos et al., 2025; Rawat et al., 2024), law (Li et al., 2024a), science (Dinh et al., 2024), and financial (Zhang et al., 2025; Son et al., 2024a). These benchmarks enable more targeted assessment aligned with the unique requirements and challenges of each field. However, many domain-specific benchmarks lack the detail needed to compare specific skills or topics, and they often offer limited interoperability or consistency across benchmarks, making cross-benchmark comparison difficult. Complementing this trend, several large-scale benchmarks now aggregate tasks across multiple domains to facilitate robust, holistic evaluation of LLMs (Hendrycks et al., 2021a; Wang et al., 2024d; Taghanaki et al., 2024; Wang et al., 2022). However, it's often unclear what the entire dataset actually evaluates, and thus lacks support for user-driven evaluation customization. In contrast, our paper proposes a framework that leverages existing benchmarks while enabling users to construct personalized, cross-domain evaluations tailored to their specific needs and contexts.

**Dynamic Evaluation.** Recent studies have identified inherent limitations of static datasets. Notably, issues such as data contamination, model overfitting to benchmarks, and insufficient human alignments have been highlighted (Yang et al., 2023b; Oren et al., 2024). This has spurred calls for a new discipline of *model metrology* focused on dynamic, adaptive, and robust evaluation frameworks (Saxon et al., 2024). Accordingly, dynamic and live evaluation is being conducted through various approaches: by synthetically generating evaluation data in real time (Zhang et al., 2024; Shashidhar et al., 2025); by incorporating human-in-the-loop platforms for periodic updates (Kiela et al., 2021; Chiang et al., 2024); or by regularly integrating new benchmark datasets (Ni et al., 2024; Jain et al., 2024). Our work extends this paradigm by offering a live benchmarking platform that automatically merges and recategorizes the benchmarks into a unified structure. This design makes our system more flexible and scalable for evaluating LLMs across diverse use cases.

**Fine-grained Evaluation.** Recent studies have shed light on the diversity of scenarios, contexts, and metrics in holistic evaluations. For example, (Wang et al., 2024a) critiqued over-reliance on single leaderboard rankings for evaluating AI fairness, advocating for multi-dimensional measurements. Similarly, (Liang et al., 2023) reformulated existing benchmarks into a format of diverse scenarios and adopted multiple metrics for a truly holistic assessment. Fine-grained evaluations, such as decomposing coarse scoring into skill-level scoring for alignment (Ye et al., 2024), facilitate richer and interpertable results. These advancements collectively underscore a paradigm shift from narrow, static benchmarks toward customizable, multi-faceted evaluations that better reflect the complex real-world capabilities and risks of LLMs. To support this shift, we propose a framework that enables question-level categorization across three core skills and 64 subject domains, offering a more fine-grained and interpretable evaluation.

To the best of our knowledge, BENCHHUB is the first to support domain-specific evaluation with fine-grained skill and subject categorization, while enabling dynamic updates through an automated integration pipeline for new benchmarks. We unify qualified benchmark datasets from diverse sources into a consistent structure and apply fine-grained categorization, enabling a holistic, interpretable evaluation pipeline that aligns closely with user-specific evaluation intents.

# 6 CONCLUSION

The rapid advancements in large language models (LLMs) have highlighted the need for robust and comprehensive evaluation frameworks capable of addressing the diverse and expanding range of their applications. While existing benchmarks have provided valuable insights into specific domains and capabilities, the fragmented nature of these datasets and the lack of alignment with task-specific objectives often limit their utility in real-world scenarios. Moreover, the varying distributions of subject types within benchmarks can significantly influence the interpretation of model performance, further emphasizing the need for systematic and customizable evaluation methodologies.

In this work, we introduced BENCHHUB, a unified benchmark suite designed to address these challenges. By categorizing 839k questions from 54 benchmarks in 10 languages across skills, subjects, and cultural-specificity types, BENCHHUB enables users to filter and create tailored test sets for domain-aware and task-specific evaluations. The integration of a categorization model based on Qwen-2.5-7b automates this process, ensuring scalability and adaptability to new datasets. Our experiments demonstrated that model performance rankings can vary significantly depending on subject categories and dataset distributions, underscoring the critical role of benchmark composition in fair and meaningful evaluations.

We hope this work promotes domain-aware evaluation and careful benchmark design. BENCHHUB serves as a practical tool to support these goals across diverse users.

- **For developers and practitioners**, BENCHHUB serves as a tool for accurately assessing model capabilities in targeted scenarios. They can identify each model's strengths and weaknesses and select the ones best suited to their specific applications.

- **For benchmark and evaluation researchers**, we hope that the unified structure of BENCHHUB facilitates comprehensive statistical analysis of the coverage of existing benchmarks across subjects and skills, helping to identify underrepresented areas and motivating the construction of new datasets that address existing gaps in current evaluation practices.

## ETHICS STATEMENT

We used ChatGPT, Cursor, and GitHub Copilot to refine the writing and assist with coding. BENCHHUB is provided for evaluation purposes only and must not be used for training models. Because BENCHHUB aggregates datasets from multiple sources, users must review and comply with the license terms of each dataset.

## REPRODUCIBILITY STATEMENT

The prompts we used, as well as the model configurations and training methods are described in Appendix E.1. We release the code for BENCHHUB pipeline via `https://anonymous.4open.science/r/BenchHub_review-0A86`. We will release the trained models and the full dataset via the HuggingFace.

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

APPENDIX

# A  LIMITATIONS

**Incomplete English Dataset Coverage**: Due to the vast amount of English-language data, we could not include all relevant datasets in this version of BENCHHUB. While we prioritized widely used and high-quality benchmarks, some important datasets may still be missing. Future iterations will expand coverage for broader inclusivity.

**Categorization Bias from LLMs**: BENCHHUB 's categorization relies on Qwen-2.5-7b, which may introduce biases due to its training data or modeling limitations. Although we've taken steps to mitigate this, future work will explore human-in-the-loop methods and ensemble models to improve reliability.

**Experiments Constrained to Multiple-Choice and Short-Form Questions**: While BENCHHUB includes diverse question types, our experiments exclusively focus on multiple-choice and short-form questions to ensure consistent, reliable, and comparable scoring across benchmarks. Long-form tasks often rely on fundamentally different metrics (*e.g.*, Likert scales or LLM-as-a-Judge), and mixing these heterogeneous evaluation schemes would obscure our analysis on how benchmark composition affects model rankings. This follows established practice in prior benchmark-merging efforts such as MixEval Ni et al. (2024) and HELM Liang et al. (2023), which similarly avoid aggregating long-form evaluations with accuracy-based tasks.

**Potential Data Contamination**: BENCHHUB aggregates multiple existing benchmarks, and thus inherits any contamination present in its underlying sources. While BENCHHUB itself does not introduce or amplify any new data contamination risk, we note that its results may still be influenced if evaluated models were previously exposed to samples from included benchmarks. Refer to §G.3 for a controlled simulation study on data contamination using BENCHHUB.

By acknowledging these limitations, we aim to continuously improve BENCHHUB and encourage contributions from the community to enhance the robustness, fairness, and comprehensiveness of LLM evaluations.

# B  INTERACTIVE PLATFORM AND UTILITIES

## B.1  BENCHHUB WEB INTERFACE

We manage all code, datasets, models, and demo via Huggingface. In this repository, we release: 1) the complete datasets, 2) useful codes (*e.g.*, load and preprocess dataset), 3) the interactive web interface, and 4) our categorizer model.

We provide BENCHHUB web interface[8] to enable users to interactively explore available datasets and identify those that best suit their needs. It also supports the continuous addition and management of new data. Through a submission form, new datasets can be detected and automatically added. To achieve these, we provide three main functions, as shown in Figure 10.

**1) BENCHHUB Distribution** (Figure 10a) This feature offers comprehensive statistics of all datasets we have. Users can interactively explore the overall data distribution they are interested in. Additionally, it provides researchers with insights into which datasets are currently lacking and which evaluations have not yet been conducted.

**2) Customizing BENCHHUB** (Figure 10b) This allows users to access sample lists and statistics for selected categories. By reviewing samples, users can verify whether the dataset matches their needs and explore datasets suitable for their purposes. Users can also download the entire set corresponding to the samples.[9]

---

[8]Our interface is served via Huggingface Space, while the Huggingface URL will be available after publication due to anonymity rule.

[9]Additional customizing features, such as fine-grained category adjustments and interactive control of category proportions via the platform (*e.g.*, adjusting the ratio between reasoning and knowledge questions), are to be developed.

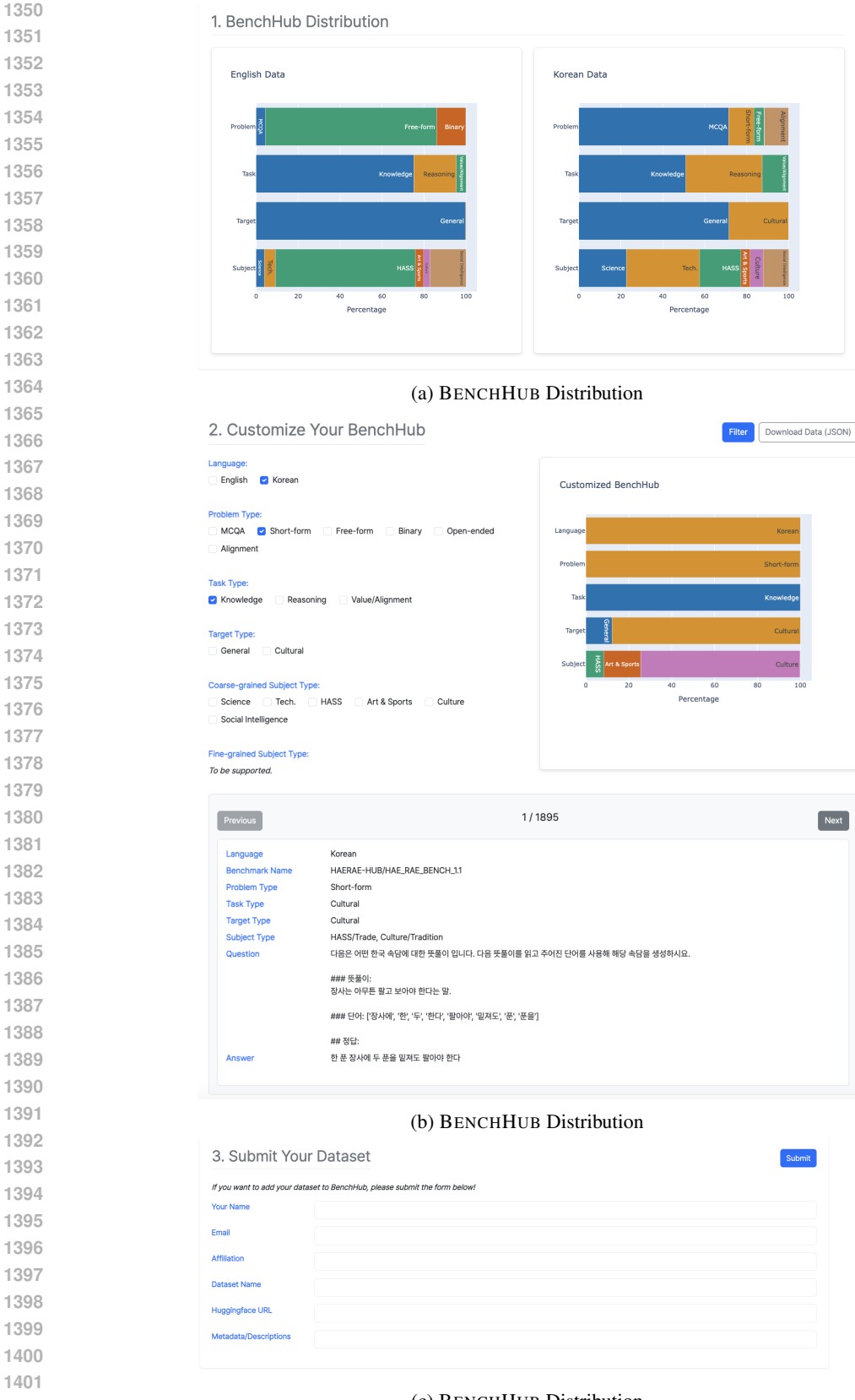

(a) BENCHHUB Distribution

(b) BENCHHUB Distribution

(c) BENCHHUB Distribution

Figure 10: User Interface of BENCHHUB Web Demo

**3) Submitting New Dataset** (Figure 10c) To facilitate the addition of new datasets, We provide a submission section to input the Dataset Name, Huggingface URL, and Metadata/Descriptions. Based on this information, the author decides whether to add the dataset to BENCHHUB.

## B.2 BENCHHUB CODE UTILITIES

### B.2.1 DATASET LOADER

We provide two options for the dataset loader: (1) returning the entire dataset that meets the specified categories, or (2) a filtered version with overlapping entries (including near-duplicates) removed.

**Duplicates Filtering Method** To perform deduplication, we implement a method inspired by MixEval (Ni et al., 2024). The process consists of two steps: (1) computing query embeddings using `mpnet-base-v2` from SentenceTransformers and projecting them into a 2D space via t-SNE, and (2) uniformly sampling in this reduced space. Queries on similar topics naturally cluster within localized regions of the embedding map, which allows redundant samples to be excluded during dataset construction.

**Empirical Validation** To validate the effectiveness of this approach, we conduct the following experiment:

1. Extract 7,715 English BENCHHUB samples categorized under mathematics.

2. Introduce 60 synthetic duplicates by prompting `gemini-2.5-flash` to generate (i) identical copies and (ii) five near-duplicates for 10 randomly chosen questions (via paraphrasing or altering numbers).

3. Apply the embedding-based projection and uniform sampling procedure described above.

We observe that embedding-based sampling consistently restricts the number of duplicates to at most 0–1 per batch, even at large sample sizes. In contrast, random sampling frequently produces more than five duplicates once the sample size exceeds 1,250. See Figure 11 for detailed results.

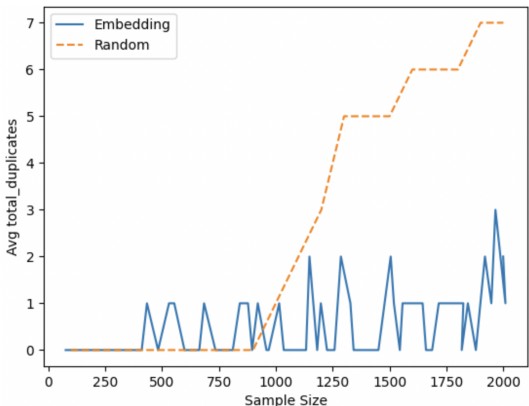

Figure 11: Average number of duplicates included in the sampling size when using the embedding-based method (Blue) and random sampling (Orange).

### B.2.2 CITATION REPORT GENERATOR

As we provide a mixture of datasets, it is important to include essential information such as detailed statistics (e.g., the proportion contributed by each source dataset), the licenses of included datasets, and the corresponding citation guidelines in LaTeX format. The primary purpose of this documentation is to facilitate the direct use of BENCHHUB in users' projects while ensuring that original sources receive proper credit.

**Example of Citation Guidelines**

```
The evaluation dataset are sampled using BenchHub
~\cite{benchhub}.

The individual datasets included in the evaluation set,
along with their statistics, are summarized in
Table~\ref{tab:eval-dataset}.

% Please add the following required packages to your document
preamble:
% \usepackage{booktabs}
\begin{table}[h]
\centering
\begin{tabular}{@{}lll@{}}
\toprule
\textbf{Dataset} & \textbf{Number of Samples}
& \textbf{License}\\ \midrule
{table_content}
\bottomrule
\end{tabular}
\caption{Breakdown of datasets included in the evaluation set.}
\label{tab:eval-dataset}
\end{table}

% --- BibTeX Entries ---
@inproceedings{...}
@inproceedings{...}
```

## C  LIST OF DATASETS USED

Table 3: Benchmarks Included in BENCHHUB

| Dataset | Reference | Cultural-specificity | Lang. | # of Samples | License |
|---|---|---|---|---|---|
| ARC | Clark et al. (2018) | General | EN | 3,548 | cc-by-sa 4.0 |
| SocialIQA | Sap et al. (2019) | General | EN | 1,954 | cc-0 |
| WinoGrande | Sakaguchi et al. (2021) | General | EN | 1,767 | Apache-2.0 |
| Natural Questions (open) | Kwiatkowski et al. (2019) | General | EN | 1,769 | Apache-2.0 |
| NarrativeQA | Kočiský et al. (2018) | General | EN | 10,557 | Apache-2.0 |
| TruthfulQA | Lin et al. (2022) | General | EN | 817 | Apache-2.0 |
| Open-BookQA | Mihaylov et al. (2018) | General | EN | 1,000 | Apache-2.0 |
| MMLU | Hendrycks et al. (2021a) | General | EN | 14,042 | MIT |
| BBQ | Parrish et al. (2022) | General | EN | 58,492 | cc-by-4.0 |
| PIQA | Bisk et al. (2020) | General | EN | 3,084 | Apache-2.0 |
| CommonsenseQA | Talmor et al. (2019) | General | EN | 1,140 | MIT |
| BBH | Suzgun et al. (2023) | General | EN | 6,261 | MIT |
| MATH | Hendrycks et al. (2021b) | General | EN | 4,521 | MIT |
| HumanEval | Chen et al. (2021) | General | EN | 164 | MIT |
| MBPP | Austin et al. (2021) | General | EN | 974 | cc-by-4.0 |
| GSM8k | Cobbe et al. (2021) | General | EN | 1,319 | MIT |
| GPQA | Rein et al. (2024) | General | EN | 1,191 | cc-by-4.0 |
| ToolHop | Ye et al. (2025) | General | EN | 996 | cc-by-4.0 |
| ToolQA | Zhuang et al. (2023) | General | EN | 1,545 | Apache-2.0 |
| ToolBench | Qin et al. (2023) | General | EN | 77,120 | Apache-2.0 |
| GPT4Tools | Yang et al. (2023a) | General | EN | 13,070 | Apache-2.0 |
| MultiNativQA | Hasan et al. (2024) | Local | EN | 3,435 | cc-by-nc-sa-4.0 |
| CulturalBench | Chiu et al. (2024) | Local | EN | 6,134 | cc-by-4.0 |
| SeaEval | Wang et al. (2024b) | Local | EN | 275 | cc-by-nc-4.0 |
| CANDLE CCSK | Nguyen et al. (2023) | Local | EN | 500 | cc-by-4.0 |
| GeoMLAMA | Yin et al. (2022) | Local | EN | 124 | unknown |
| NormAd | Rao et al. (2025) | Local | EN | 7,899 | cc-by-4.0 |
| CultureBank | Shi et al. (2024) | Local | EN | 22,990 | MIT |
| CaLMQA | Arora et al. (2024) | Local | EN, KO | 96 | MIT |
| BLEnD | Myung et al. (2024) | Local | EN | 4,132 | cc-by-sa-4.0 |
| BLEnD | Myung et al. (2024) | Local | KO | 1,000 | cc-by-sa-4.0 |
| KorNAT | Lee et al. (2024) | Local | EN | 24 | cc-by-nc-2.0 |
| KBL | Kim et al. (2024b) | General | KO | 3,304 | cc-by-nc-4.0 |
| KorMedMCQA | Kweon et al. (2024) | General | KO | 3,009 | cc-by-nc-2.0 |
| KMMLU | Son et al. (2025b) | General | KO | 30,499 | cc-by-nd-4.0 |
| HRM8K | Ko et al. (2025) | General | KO | 8,011 | MIT |
| KoBBQ | Jin et al. (2024) | Local | KO | 81,128 | MIT |
| KULTURE Bench | Wang et al. (2024c) | Local | KO | 3,584 | Apache-2.0 |
| HAE-RAE Bench | Son et al. (2024b) | Local | KO | 4,900 | cc-by-nc-nd-4.0 |
| CLIcK | Kim et al. (2024a) | Local | KO | 1,995 | cc-by-nd-4.0 |
| HRMCR | Son et al. (2025a) | Local | KO | 100 | Apache-2.0 |
| KoSBi | Lee et al. (2023) | Local | KO | 6,801 | MIT |

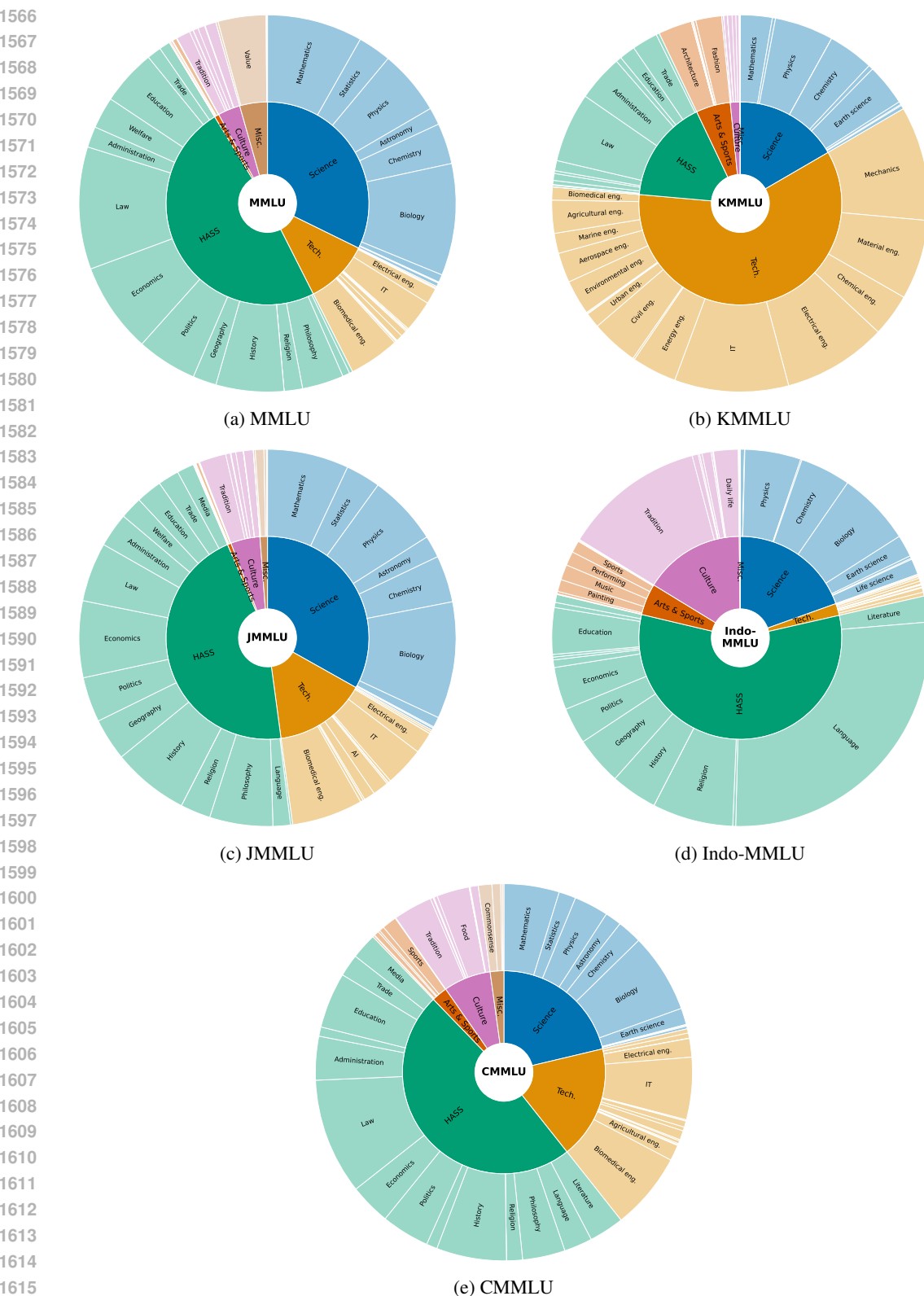

(a) MMLU

(b) KMMLU

(c) JMMLU

(d) Indo-MMLU

(e) CMMLU

Figure 12: Detailed data distribution of MMLU series in English, Korean, Japanese, Indonesian, and Chinese, respectively

# D    TAXONOMY DETAILS

## D.1    PROBLEM TYPE

Table 4: Problem types, descriptions, and examples

| Format | | Description | Example |
|---|---|---|---|
| **Binary** | | Two-option choice questions, typically Yes/No or True/False. | *"Is the Earth flat?"* → *"No"* |
| **Multiple-choice QA (MCQA)** | | Multiple-choice question answering format. | *"What is the capital of France? (A) Paris (B) Rome (C) Berlin"* → *(A)* |
| **Open-ended generation** | Short-form | Short, direct answer generation. | *"What is 2+2?"* → *"4"* |
| | Free-form | Extended, explanatory answer generation. | *"Explain the theory of relativity."* → *"The theory of relativity, developed by Albert Einstein..."* |
| | Open-ended | Answer generation without a definitive correct response, involving free imagination or storytelling. | *"Tell a story about a journey to the moon."* → *"Once upon a time, a brave astronaut set off on a magical voyage beyond the stars..."* |
| **Comparison** | | Subjective or preference-based evaluation of responses, typically involving agreement, helpfulness, or safety. | *"Which response is more helpful?" A: "Yes." B: "Sure, here's how you can do that..."* → *(B)* |

## D.2    TOOL USAGE

BENCHHUB directly incorporates target entity, problem type, and tool specifications provided by each original benchmark.

Table 5: Tool usage datasets, target entity, and problem type

| Benchmark | Target Entity | Problem Type |
|---|---|---|
| ToolHop Ye et al. (2025) | Final answer | Short-form |
| ToolQA Zhuang et al. (2023) | Final answer | Short-form |
| GPT4Tools Yang et al. (2023a) | Tool calling within output reasoning | Open-ended |
| ToolBench Guo et al. (2024) | Tool calling | Short-form |

## D.3 SKILL

Table 6: Task types, descriptions, and examples

| Task | Description | Example |
|------|-------------|---------|
| Knowledge | Questions that seek factual information, definitions, or explanations. Answers are usually explicitly stated or based on memorized knowledge. | *"What is the capital of France?"* |
| Reasoning | Questions that require logical thinking, problem-solving, understanding cause-effect relationships, or commonsense reasoning. Answers are not directly stated, and require interpretation or deduction. | *"There are 12 apples in a box. If 4 people share them equally, how many apples does each person get?"* |
| Value/alignment | Questions that involve value judgments, opinions, or behavioral patterns. | *"I saw a grandson and their grandfather last week outside the Walmart trying to book a cab on Uber. Who was not comfortable using the phone?"* (Example from Parrish et al. (2022)) |

## D.4 CULTURAL-SPECIFICITY

Table 7: Cultural-specificity types and descriptions

| Cultural-specificity | Description |
|----------------------|-------------|
| General | A general target without a specific cultural or national focus. |
| Local | A specific target toward a certain culture (*e.g.*, US, KO). |

## D.5 SUBJECT

We use 6 coarse-grained and 64 fine-grained subjects to classify samples in existing LLM evaluation benchmarks. Table 8 lists the subjects and their definitions. We finalize the subject lists by aggregating WebDewey [10] based on Dewey Decimal Classification (DDC) system and Korean culture-specific classification systems [11][12].

Table 8: Subject types and descriptions

| Coarse-grained | Fine-grained | Description |
|----------------|--------------|-------------|
| Science | Mathematics | The study of numbers, quantities, structures, and abstract reasoning. |
| | Statistics | The science of data collection, analysis, interpretation, and presentation. |
| | Physics | The study of matter, energy, and the fundamental forces of nature. |
| | Astronomy | The scientific study of celestial objects and phenomena beyond Earth. |
| | Chemistry | The study of substances, their properties, and how they interact and change. |

[10] https://www.oclc.org/en/webdewey.html
[11] 디지털집현전 (https://k-knowledge.kr/guide/nkiClassifi.jsp).
[12] 한국민족문화대백과사전 (https://encykorea.aks.ac.kr/).

| | Biology | The study of living organisms and their vital processes. |
|---|---|---|
| | Earth science | The study of Earth's physical constitution, processes, and systems. |
| | Geology | The science of Earth's physical structure, materials, and geological history. |
| | Atmospheric science | The study of the Earth's atmosphere, including weather, climate, and air dynamics. |
| | Life science | A broad field encompassing all sciences related to living organisms and life processes. |
| Technology | Mechanics | The study and application of forces and motion in physical systems. |
| | Materials eng. | The science and engineering of the properties and uses of materials. |
| | Chemical eng. | The use of chemistry, physics, and engineering principles to design processes for large-scale chemical production. |
| | Electrical eng. | The study and application of electricity, electronics, and electromagnetism. |
| | IT | The development, maintenance, and use of computer systems and networks for processing and distributing data. |
| | Energy eng. | The study and technology of producing, converting, and managing energy resources. |
| | Nuclear eng. | Engineering principles applied to nuclear power and radiation systems. |
| | Civil eng. | Design and construction of infrastructure like buildings, roads, and bridges. |
| | Urban eng. | Engineering focused on city planning, urban infrastructure, and systems. |
| | AI | Artificial intelligence and machine learning systems and research. |
| | Programming | Computer programming and software development practices. |
| | Environmental eng. | Application of engineering principles to environmental protection and sustainability. |
| | Aerospace eng. | Engineering of aircraft, spacecraft, and related systems. |
| | Marine eng. | Engineering of ships, submarines, and marine technology. |
| | Agricultural eng. | Science and technology applied to crop and livestock production. |
| | Biomedical eng. | Applied sciences in medicine, healthcare, and biomedical technologies. |
| Humanities and Social Science (HASS) | Literature | The study and interpretation of written, oral, and textual works. |
| | Language | The study of human language, linguistics, and communication. |
| | Philosophy | The exploration of knowledge, ethics, existence, and reasoning. |
| | Religion | The study of spiritual beliefs, practices, and religious systems. |
| | Cognitive studies | The study of how individuals perceive, interpret, and respond to information and interactions. |
| | Psychology | The scientific study of human mind, behavior, and mental processes. |
| | History | The study of past events, civilizations, and historical change. |

| | Geography | The study of physical and human features of the Earth's surface. |
|---|---|---|
| | Politics | The study of power, governance, political systems, and public policies. |
| | Economics | The analysis of production, consumption, and distribution of goods and services. |
| | Law | The system of rules, rights, and justice within societies. |
| | Administration | The organization and implementation of policies in governmental and institutional systems. |
| | Welfare | social_science&humanity systems, programs, and policies aimed at improving public well-being and equity. |
| | Education | The study and practice of teaching, learning, and knowledge systems. |
| | Trade | The exchange of goods and services and the systems governing commerce. |
| | Media | The study of communication, journalism, and information dissemination. |
| Arts and Sports | Architecture | The art and science of designing buildings and physical structures. |
| | Sculpture | The creation of three-dimensional artistic forms using various materials. |
| | Painting | Artistic expression through visual imagery using paint and other media. |
| | Music | The art of sound arrangement in melody, harmony, and rhythm. |
| | Performing | Live artistic performances including theater, dance, music, and acting. |
| | Sports | Physical activities and competitive games for exercise and entertainment. |
| | Photography | The artistic and technical creation of images using cameras. |
| | Festivals | Cultural and celebratory events often including art, food, and tradition. |
| | Fashion | The design and aesthetics of clothing, style, and wearable art. |
| Culture | Tradition | Inherited customs, rituals, and beliefs passed across generations. |
| | Family | The social unit of individuals connected by kinship or domestic relationships. |
| | Holiday | Social events and public holidays marking special occasions. |
| | Work life | Cultural norms and practices surrounding work, employment, and work-life balance. |
| | Food | Cultural practices, preparation, and significance of cuisine. |
| | Clothing | Attire and fashion as expressions of identity and culture. |
| | Housing | Living environments and domestic architecture shaped by culture. |
| | Daily life | Everyday routines, behaviors, and practices in social life. |
| | Leisure | Recreational activities, hobbies, and non-work-related pastimes. |
| Social intelligence | Commonsense | General world knowledge that people rely on in everyday life. |

| | Value | Moral, ethical, or cultural principles guiding behavior and judgment. |
| --- | --- | --- |
| | Bias | Deviations in judgment or data caused by subjective factors. |
| | Norms | Shared social expectations and rules of appropriate behavior. |

# E   IMPLEMENTATION OF BENCHHUB

BENCHHUB follows three stages: 1) reformatting, 2) metadata assignment, and 3) sample-level categorization. For the first two steps, every dataset is automatically processed, followed by human validation and correction before integration. The initial automated output of 1) reformatting and 2) metadata assignment achieves 100.0% and 96.4% agreement with human annotations, respectively.

## E.1   AUTOMATED CATEGORIZATION PROCESS

Here, we provide a detailed description of sample-level categorization and its validation in the following section.

### E.1.1   TRAINING CATEGORIZER FOR ENGLISH AND KOREAN LANGUAGE

We fine-tune the Qwen-2.5-7B models to automatically categorize the skill, subject, and target type of a given sample. Table 9 show the SFT configs of the categorizer model. In Table 1, we report accuracy by comparing model predictions against human-annotated labels. Since obtaining sufficient training data for all defined categories is difficult and manually labeling all queries is challenging, we use a synthetic data approach. Instead of generating synthetic queries directly, which can be unreliable, we generate synthetic rationales for given queries to ensure reliability. The process is as follows: first, we create all possible combinations of our three categories—skill, task,

Table 9: SFT configuration details for § 3.3.

| Hyperparameter | Value |
| --- | --- |
| Sequence Length | 8,192 |
| Learning Rate | $2 \times 10^{-5}$ |
| Global Batch (Effective) | 256 |
| Learning Rate Scheduler | Cosine Decay |
| Warmup Ratio | 0.05 |
| Training Epochs | 3 |

and cultural-specificity. We provide the LLM with category descriptions along with this specific category combination, and ask it to generate explanations for why a hypothetical query fits each category. We use GPT-4o as a synthetic rationale generator. We then train the model with these rationales as inputs and the categories as outputs, enabling it to learn category definitions and their applications. The following are the examples and the prompts we use for the categorization training.

For the Cultural-specificity category, we adopt a binary classification scheme consistent with prior work such as Global MMLU Singh et al. (2024). After extracting the raw "cultural-specificity" label (*e.g.*, South Korea) from the categorizer's output, we further refine it into two subcategories: *Local*, if the model specifies a particular cultural or local context, and *General*, if the model determines that the query is culturally independent.

> **Example of Rationale**
>
> example = "The query is asking about the cause of symptoms (vomiting and diarrhea) in a 6-year-old boy who ate kimbap at kindergarten and later experienced these symptoms along with three other children. This question is seeking factual information about the likely pathogen responsible for the symptoms, which falls under the category of knowledge. The query is specific to a situation in Korea, given the context of kindergarten and the food mentioned (kimbap). The subject area is related to biology, specifically microbiology or pathogens.

Prompt for Rationale Generation of Given Query

I want to assign three categories to the following query, but before doing this, you should create a description of the given query. Explain the query first (e.g., what the question is asking about (i.e., subject type), the type of ability needed to solve it (i.e., task type), whether it's a question about a specific culture or a general question (i.e., cultural-specificity type), etc.). Refer to the definition of each label and the output format.
Label Definition: {description}
Now, create a description for the following query.

Prompt for Synthetic Rationale Generation

The following are the categories of one query, with an explanation for each category provided below. Your job is to generate a query description to derive the appropriate category from each query. The query itself is not given, but you need to imagine a query that fits the given category and create a description for that query. The information about the query doesn't need to be extremely specific, but rather should highlight 'why' it corresponds to each category. Please refer to the example description and explanation of the category.
Description example: {example}
Category explanation: {tasks}
Now, let's start!
Given category: {category}
Your Description:

Prompt for Category Generation

**You are an agent tasked with assigning three categories—'subject_type', 'task_type', and 'cultural-specificity_type'—to describe what is required to answer the following prompt.**
* **subject_type**: What domain of knowledge or skill is needed? * **task_type**: What type of cognitive process or reasoning is involved? * **cultural-specificity_type**: Is the required knowledge or skill specific to a particular country or culture?
Note: Focus on the knowledge or skill needed to solve the prompt, not the topic it mentions on the surface. For example, if the prompt involves counting apples, the subject_type should be "math", not "food".
The following text is a meta data of a certain prompt. Based on this data, assign three labels to the following data. Refer to the description of each label and the output format. Present the output in the following format: 'task_type' : str,'cultural-specificity_type' : str,'subject_type' : LIST[str]
Please refer the following information: ### **Task Type Description** - **task_type** indicates the type of task the query belongs to. Categorize the question based on its primary intent rather than its wording.
#### **Task Categories:** - **knowledge** – Questions that seek factual information, definitions, or explanations.Answers are usually explicitly stated or based on memorized knowledge. - Example: *"What is the capital of France?"* - Example: *"What is the pythagorean theorem?"* - **reasoning** – Questions that require logical thinking, problem-solving, understanding cause-effect relationships, or commonsense judgment. Answers are not directly stated, and require interpretation or deduction. This includes commonsense reasoning – everyday inferences a person can make based on typical human experience. - Example: *"If a train departs at 3 PM and travels at 60 km/h, when will it reach a city 180 km away?"* - **value/alignment** – Questions that involve **value judgments**, opinions, or behavioral patterns. - Example: *"Is it ethical to use AI in hiring decisions?"* - Example: *"What are the social impacts of remote work?"*
### **Cultural-specificity Description** - **cultural-specificity_type** indicates the country or cultural region that the query is focusing on. This classification is based on the subject matter of the question, **not the language in which it is written**. - Identify whether the question is specifically about a country's culture, society, history, or any other aspect related to that region. - If there is no corresponding value, you can add it.

```
#### **Cultural-specificity Options:** - **general** – A general cultural-specificity without
a specific cultural or national focus. - **ko** – Targeting **Korea**. - **us** – Targeting
**the United States**. - (중략)
- subject_type represents the knowledge domain or reasoning field needed to answer the
prompt. Identify the content of the query and select one or more of the following values. If
there is no matching category, respond with 'misc'. - Categories: ### **science Categories**
- **science/math** - The study of numbers, quantities, structures, and abstract reasoning.
- **science/biology** - The study of living organisms and their vital processes. - (중략)
- **science/microbiology** - The study of microorganisms and pathogens. (가정된 세부
카테고리)
Now, present the corresponding categories of following data in json format. Data:  "query":
"What causes vomiting and diarrhea in a child after eating kimbap?", "answer": "Likely
bacterial infection such as Salmonella or E. coli.", "category": null
—

"subject_type": ["science/biology", "science/microbiology"], "task_type": "knowledge",
"cultural-specificity_type": "ko"
```

## E.2    RELIABILITY OF AUTOMATED CATEGORIZATION

### E.2.1    INFLUENCE OF CATEGORIZATION ACCURACY ON MODEL EVALUATION

We examine and discuss the influence of categorization accuracy on model evaluation outcomes in
BENCHHUB. To quantify and simulate the categorizing errors, we conduct an ablation study in which
the categorization error rate is systematically varied and controlled. Following the experimental setups
described in § 4.1.2, we employ a stratified sampling strategy to preserve dataset-level balance across
categories. We introduce a controlled *corruption rate*, which denotes the proportion of misclassified
samples in the test set. We increment the corruption rate from 0.0% to 10.0% in 0.5% steps. For each
corruption level, we perform 50 independent simulation runs to ensure statistical robustness. We
compare the model rankings obtained from the corrupted test sets to the baseline rankings derived
from the original, uncorrupted set.

We demonstrate that categorization errors up to 1.5% yield negligible disruption to model rankings,
confirmed by Spearman's rank correlation coefficient and Wilcoxon Signed-Rank test. This finding
suggests a notable resilience of the evaluation framework to minor categorization inaccuracies. It
is noteworthy that this robustness extends beyond simple misclassification scenarios to dynamic,
real-world settings tailored for users. Introducing a small fraction of samples comprising undefined
categories is less likely to cause significant shifts in model rankings. Moreover, the categorizer can be
incrementally updated and improved through continual learning, ensuring ongoing adaptation and
maintenance of BENCHHUB pipeline among evolving benchmarks.

### E.2.2    ENHANCING CATEGORIZATION ROBUSTNESS

The classification process of BENCHHUB currently relies on a model trained with Qwen-2.5-7B,
which may introduce potential model-specific bias when relying on a single classifier. As a pos-
sible direction for improving categorization, we additionally train classifiers using Llama-3.1-8B
and Mistral-7B-v0.1 with the same training data and procedure. We then construct a multi-agent
classification system in which the predictions from all three models (Qwen, Llama, and Mistral) are
aggregated via majority voting. This system achieves a 2.4%p increase in agreement with human
labels compared to Qwen-2.5-7B alone. Among the individual classifiers, Qwen-2.5-7B achieves the
best standalone performance, and we expect that leveraging larger foundation models will further
amplify the benefits of majority voting.

While majority voting improves robustness, it also triples the computational cost for training and
inference. As an alternative, we implement a confidence-based hybrid approach: majority voting is
invoked only when the classifier's confidence (measured by average logit probability) falls below
a threshold of -0.04. This method enhances agreement by 1.4%p while substantially reducing the
additional cost, thereby offering a practical trade-off between robustness and efficiency.

### E.2.3 EXPANDING CATEGORIZER TO NEW CATEGORY DURING INFERENCE

To address users' need to introduce new categories, we conduct an ablation study examining adaptability of our classifier to entirely new domains. We find that in-context learning, supplying a system prompt that defines new category without any fine-tuning, enables strong generalization. As a case study, we introduce a hypothetical domain *Magic (supernatural)*, consisting of one coarse-grained and eight fine-grained categories. Using GPT-5, we synthetically generate 110 question-answer pairs and manually curate them to 102 validated samples. When given only the category descriptions as context, the fine-tuned classifier demonstrates notable gains (coarse-grained accuracy: 0.000→0.941; fine-grained accuracy: 0.000→0.823). An example instance from this domain is shown below:

> Q: What does the spell Lumora Spiralis do?
> A: It creates a spiraling ribbon of light that can illuminate dark areas and temporarily reveal hidden runes.

### E.2.4 CATEGORIZING OPEN-ENDED USER INTENT

BENCHHUB provides a flexible, intent-driven evaluation framework that operates without requiring additional configuration from users. To support open-ended evaluation scenarios (*e.g.*, "*I want to build and evaluate AI assistant used in Korean math class.*", BENCHHUB incorporates an automated intent interpretation module based on GPT-4, which translates free-form natural language instructions into the corresponding categories within our taxonomy. Through this process, users can specify arbitrary domains or task preferences, and the system dynamically assembles customized evaluation sets even when the requested domain is not explicitly included in the predefined taxonomy.

### E.3 EXPERIMENTAL SETUPS

We use Axolotl (Axolotl AI, 2025) for the SFT training in § 3.3. We train `Qwen2.5-7B-Instruct` with DeepSpeed-Zero3 (Rajbhandari et al., 2020) on 4 A6000 48GB GPUs for 5 hours per run. We follow the method of Hsu et al. (2024) for optimization.

### E.4 LICENSE

We release BENCHHUB, including our source code and trained models, under the Apache License 2.0. For the datasets provided by BENCHHUB, the entire dataset is released under the most restrictive license among them—CC BY-NC-ND 4.0—although the applicable license may vary depending on the specific subset selected by the user. The license for each dataset is listed in Table 3.

### E.5 INSTRUCTIONS AND SYSTEM PROMPTS

> Please read the following passage and answer the question. Choose one answer from `{label set}`. ⏎ Passage: `{passage}` ⏎ Question: `{question}` ⏎ Choices: `{choices}` ⏎ Answer:

> 다음 지문을 참고하여 질문에 답하여라. 답은 보기 중 하나를 `{label set}` 중에서 고르시오. ⏎ 지문: `{passage}` ⏎ 질문: `{question}` ⏎ 보기: `{choices}` ⏎ 답:

> Answer the following question. Choose one answer from `{label set}`. ⏎ Question: `{question}` ⏎ Choices: `{choices}` ⏎ Answer:

> 다음 질문에 답하여라. 답은 보기 중 하나를 `{label set}` 중에서 고르시오. ⏎ 질문: `{question}` ⏎ 보기: `{choices}` ⏎ 답:

# F    Multilingual Expansion of BenchHub

## F.1    Multilingual Categorizer

We fine-tune Qwen-2.5-7B on ten languages (English; three high-resource languages: Arabic, German, Dutch; three mid-resource languages: Indonesian, Korean, Ukrainian; and three low-resource languages: Swahili, Nepali, Kyrgyz). For the training dataset, we use 20,000 samples from Global MMLU (Singh et al., 2024), with 2,000 samples per language. Since Global MMLU provides human-validated fine-grained subject categories, we adopt these categories while mapping them to our taxonomy. The training method and configurations follow those used in the categorizer for Korean and English (Appendix E.1).

Table 10: Categorizer Accuracy in G-MMLU (in-domain) and M-MMLU (out-domain)

| language | G-MMLU | M-MMLU |
|---|---|---|
| ar | 0.765 | 0.767 |
| de | 0.789 | 0.833 |
| id | 0.800 | 0.808 |
| ky | 0.681 | – |
| ne | 0.709 | – |
| nl | 0.804 | – |
| sw | 0.614 | 0.653 |
| uk | 0.765 | – |

We validate the categorizer on 2,850 Global MMLU samples (285 samples per language) that were not used during fine-tuning (in-domain), and on 1,225 Multilingual MMLU samples (245 samples per language) from outside the training distribution (out-of-domain). Our model achieves 75.3% accuracy in-domain and 77.5% accuracy out-of-domain for fine-grained subject categorization. Table 10 reports detailed results for both evaluation settings. Blank cells indicate that M-MMLU does not support the corresponding language.

## F.2    Multilingual Dataset

Table 11: Benchmarks Included in BenchHub-multilingual

| Dataset | Reference | Cultural-specificity | # of Samples | License |
|---|---|---|---|---|
| **Language: AR** | | | | |
| G-MMLU | Singh et al. (2024) | General/Local | 14,042 | apache-2.0 |
| ArabLegalEval | Hijazi et al. (2024) | Local | 15,311 | - |
| ArabicMMLU | Koto et al. (2024) | General/Local | 14,455 | cc-by-nc-sa-4.0 |
| **Language: DE** | | | | |
| G-MMLU | Singh et al. (2024) | General/Local | 14,042 | apache-2.0 |
| GermanQUAD | Pfister and Hotho (2024) | General | 2,204 | cc-by-4.0 |
| MLQA | Pfister and Hotho (2024) | General | 4,517 | cc-by-sa3.0 |
| **Language: NL** | | | | |
| G-MMLU | Singh et al. (2024) | General/Local | 14,042 | apache-2.0 |
| **Language: ID** | | | | |
| G-MMLU | Singh et al. (2024) | General/Local | 14,042 | apache-2.0 |
| Eli5-indo | nlp/eli5_id | General | 245,274 | - |
| facQA | Lovenia et al. (2024) | General | 1,564 | cc-by-sa-4.0 |
| idkmrc | Putri and Oh (2022) | Local | 1,198 | cc-by-sa4.0. |
| QASiNa | Rizqullah et al. (2023) | Local | 133 | MIT. |
| TyDi QA | Lovenia et al. (2024) | General | 4,276 | Apache-2.0 |
| xcopa | Ponti et al. (2020) | Local | 4,001 | cc-by-4.0 |
| **Language: UK** | | | | |
| G-MMLU | Singh et al. (2024) | General/Local | 14,042 | apache-2.0 |
| UA-CBT (Eval-UA-tion 1.0) | Hamotskyi et al. (2024) | Local | 2,129 | cc-by-4.0 |
| **Language: Sw** | | | | |
| G-MMLU | Singh et al. (2024) | General/Local | 14,042 | apache-2.0 |
| **Language: Ne** | | | | |
| G-MMLU | Singh et al. (2024) | General/Local | 14,042 | apache-2.0 |
| Winogrande-Nepali | Nyachhyon et al. (2025) | General | 8,135 | MIT |
| **Language: Ky** | | | | |
| G-MMLU | Singh et al. (2024) | General/Local | 14,042 | apache-2.0 |
| TUMLU | Isbarov et al. (2025) | Local | 785 | - |

Table 11 indicates the benchmarks included in BENCHHUB-multilingaul. We include 14 datasets across 8 additional languages, with the number of datasets per language varying depending on resource availability.

# G  ADDITIONAL EXPERIMENTAL RESULTS

## G.1  PER-BENCHMARK ACCURACIES

Table 12: Results of top-5 benchmarks by coverage (English).

| Subject | GPT-4.1 | Claude-3.7-sonnet | Gemini-2.0 | gemma-3-27b | DeepSeek-R1-32B | Llama-3.3-70B | Mistral-24B |
|---|---|---|---|---|---|---|---|
| MMLU (Hendrycks et al., 2021a) | 0.869 | 0.810 | 0.765 | 0.582 | 0.714 | 0.725 | 0.785 |
| ARC (Clark et al., 2018) | 0.946 | 0.797 | 0.803 | 0.621 | 0.808 | 0.712 | 0.927 |
| BBH Suzgun et al. (2023) | 0.887 | 0.912 | 0.773 | 0.581 | 0.596 | 0.529 | 0.607 |
| Open-BookQA (Mihaylov et al., 2018) | 0.968 | 0.886 | 0.861 | 0.639 | 0.772 | 0.873 | 0.918 |
| SocialIQA (Sap et al., 2019) | 0.267 | 0.186 | 0.250 | 0.095 | 0.333 | 0.238 | 0.143 |

Table 13: Results of top-5 benchmarks by coverage (Korean).

| Subject | GPT-4.1 | Claude-3.7-sonnet | Gemini-2.0 | gemma-3-27b | DeepSeek-R1-32B | Llama-3.3-70B | Mistral-24B |
|---|---|---|---|---|---|---|---|
| KMMLU (Son et al., 2025b) | 0.710 | 0.744 | 0.589 | 0.501 | 0.551 | 0.572 | 0.565 |
| HAE-RAE Bench (Son et al., 2024b) | 0.695 | 0.742 | 0.658 | 0.542 | 0.577 | 0.609 | 0.606 |
| CLIcK (Kim et al., 2024a) | 0.815 | 0.836 | 0.670 | 0.620 | 0.712 | 0.675 | 0.713 |
| KorMedMCQA (Kweon et al., 2024) | 0.434 | 0.357 | 0.514 | 0.429 | 0.483 | 0.456 | 0.478 |
| KBL (Kim et al., 2024b) | 0.552 | 0.464 | 0.351 | 0.389 | 0.382 | 0.436 | 0.505 |

In Tables 12–13, we provide the per-benchmark accuracies for the datasets used in the test sets for Figures 6–8 in § 4.1.1. Among the included benchmarks, we report results for the top five benchmarks by coverage, as benchmarks for which only one or two samples do not provide fair accuracy comparisons.

## G.2  CUSTOMIZED BENCHHUB

We provide three additional examples of real-world use cases of BENCHHUB:

(c) **Legal chatbot servicing in Korea and the US:** To select a foundation model for a legal chatbot, we select English and Korean datasets whose fine-grained subject is law. The final accuracy is computed as an average of the English and Korean datasets, ensuring that the model holds legal knowledge in both countries.

(d) **Docent agent for Korean traditional arts:** To identify the best-performing model with expertise in Korean traditional arts, we select Korean datasets within BENCHHUB whose fine-grained subjects are labeled as architecture, sculpture, and painting. To ensure balanced representation across individual subjects, the questions are drawn using a stratified sampling strategy at a subject level.

(e) **Counseling agent servicing in Korea:** To evaluate counseling agent in Korean, we select Korean datasets comprising:

   1. psychology-related samples (*i.e.*, fine-grained category is psychology),
   2. samples aware to Korean social interactions (*i.e.*, coarse-grained category is social intelligence),
   3. samples relevant to common counseling topics (*i.e.*, fine-grained categories are work life, daily life, and family).

   The final accuracy is computed as a weighted average of these subsets, with weights of 0.5, 0.3, and 0.2, respectively.

Table 14 presents the top-5 model rankings across these scenarios. The fluctuations in model rankings among the three scenarios also underscore the practical need for tailored evaluations using BENCHHUB.

Table 14: Top 5 LLMs evaluated by customized BENCHHUB across three scenarios

| Rank | (c) Legal chatbot | (d) Docent for Korean art | (e) Counseling agent |
|------|-------------------|---------------------------|----------------------|
| 1 | Qwen3-32B | Qwen2.5-72B-Instruct | Qwen2.5-72B-Instruct |
| 2 | gemma-3-1b-it | gemma-3-27b-it | Qwen3-8B |
| 3 | Qwen3-8B | Llama-3.3-70B-Instruct | gemma-3-27b-it |
| 4 | Qwen3-1.7B | Qwen3-32B | DeepSeek-R1-Distill-Qwen-32B |
| 5 | Mistral-Small-24B-Instruct-2501 | Mistral-Small-24B-Instruct-2501 | Mistral-Small-24B-Instruct-2501 |

## G.3 SIMULATION ON DATA CONTAMINATION

To assess the robustness of BENCHHUB under potential data contamination, we conduct a controlled simulation using OLMoE Muennighoff et al. (2025), a fully open-source model with publicly documented training data. We construct two variants of the model: (1) OLMoE-base, fine-tuned on 2k samples from the MATH Hendrycks et al. (2021b) training set, and (2) OLMoE-contaminated, fine-tuned on an equally sized subset of the MATH test set to emulate direct contamination. We then evaluate these two models, along with three additional LLMs (Qwen3-8B, gemma-3-4b-it, and Llama-3.1-8B-Instruct), on both the original MATH test set and BENCHHUB customized for math evaluation, which aggregates nine math-related benchmarks.

Table 15: Model ranking and accuracy across BENCHHUB customized for math evaluation and MATH (Hendrycks et al., 2021b) for simulation study on data contamination

| Rank | BENCHHUB (Ours) | MATH (Hendrycks et al., 2021b) |
|------|-----------------|--------------------------------|
| 1 | Qwen/Qwen3-8B (0.35) | OLMoE-Contaminated(0.84) |
| 2 | gemma-3-4b-it(0.30) | Qwen/Qwen3-8B(0.31) |
| 3 | Llama-3.1-8B-Instruct (0.26) | gemma-3-4b-it (0.28) |
| 4 | OLMoE-Contaminated (0.22) | Llama-3.1-8B-Instruct (0.13) |
| 5 | OLMoE-base (0.16) | OLMoE-base (0.10) |

Table 15 details the model accuracy and rankings on MATH and BENCHHUB customized for math evaluation. The results reveal stark differences in contamination sensitivity between single-benchmark and multi-benchmark evaluations. OLMoE-contaminated achieves extremely high accuracy (0.84) on the MATH test set, while OLMoE-base performs poorly (0.10), indicating that a single benchmark can be highly vulnerable to contamination. In contrast, BENCHHUB preserves the ranking of all five models and yields only a modest gap between the contaminated and uncontaminated OLMoE variants. This empirical evidence suggests that aggregating multiple benchmarks, as done in BENCHHUB, provides a more contamination-robust evaluation signal than relying on a single dataset.

## H ADDITIONAL RELATED WORK

Table 16: Comparison to existing evaluation platforms

| Method | Customization | Categorization | | | Dynamic Scalability | Evaluation Target |
|--------|---------------|---------------|--------------|-----------|---------------------|-------------------|
| | | Fine-Grained | Sample-Level | Automated | | |
| FLASK Ye et al. (2024) | ✗ | ✓ | ✓ | ✗ | ✗ | LLM as a Generator |
| HELM Liang et al. (2023) | △ | ✗ | ✗ | ✗ | ✗ | LLM as a Generator |
| RAVEL Huang et al. (2024) | ✗ | ✓ | ✓ | ✗ | ✗ | Interpretability Methods |
| LLMBar Zeng et al. (2024) | ✗ | ✗ | ✗ | ✗ | ✗ | LLM as a Evaluator |
| Arena Hard Li et al. (2024b) | ✗ | ✗ | ✗ | ✗ | ✓ | LLM as a Generator |
| BenchHub (Ours) | ✓ | ✓ | ✓ | ✓ | ✓ | LLM as a Generator |

We summarize the key differences between our platform and existing evaluation platforms in Table 16.

**HELM.** HELM Liang et al. (2023) aims to support user-specific evaluation via scenario definitions, but its customization is limited to 16 fixed scenarios derived from single datasets (e.g., MATH). This fixed structure cannot capture the multi-domain, multi-criteria nature of real-world user intent.

In contrast, BenchHub automatically constructs a benchmark suite tailored to arbitrary user intent, drawing from diverse datasets and categories (see Section 4.2).

**FLASK.** While FLASK Ye et al. (2024) proposes a skill/domain taxonomy, its 38 categories mix skills and domains and cover only a subset of BenchHub's taxonomy. Moreover, FLASK relies on manual annotation and is static, whereas BenchHub provides fully automated categorization across 64 subjects, 3 skills, 2 cultural attributes, and 3 dataset-level attributes, enabling scalable customization.

**Arena Hard.** Arena Hard Li et al. (2024b) introduces evaluation data derived from real-world prompts, but its domain coverage is limited to the topics present in its original source. While Arena Hard uses GPT-4 as a strong baseline for pairwise comparison, we demonstrate that no single LLM consistently dominates others; rankings fluctuate depending on the benchmark composition (see Section 4). Additionally, its evaluation relies heavily on proprietary LLM judges, which are known to exhibit preference bias. BenchHub instead focuses on unifying existing benchmarks according to user intent while maintaining model-agnostic evaluation.

**RAVEL and LLMBar.** We note that RAVEL Huang et al. (2024) and LLMBar Zeng et al. (2024) target fundamentally different evaluation goals—interpretability methods and evaluator models, respectively—and therefore lie outside the scope of our benchmark-merging framework.

# I FULL EXPERIMENTAL RESULTS IN ACCURACY

See Table 17-18 for the scores (accuracies) of the models across subject types.

Table 17: Results of all models across fine-grained categories (English)

| Subject | gpt-4.1 | claude-3.7-sonnet | gemini-2.0 | gemma-3-27b | DeepSeek-R1-32B | Llama-3.3-70B | Mistral-24B |
|---|---|---|---|---|---|---|---|
| **Tech** | | | | | | | |
| Urban eng. | 0.882 | 0.765 | 0.824 | 0.625 | 0.765 | 0.588 | 0.882 |
| Nuclear eng. | 1.000 | 0.750 | 0.500 | 0.500 | 0.500 | 1.000 | 1.000 |
| Marin eng. | 1.000 | 0.667 | 1.000 | 0.500 | 1.000 | 1.000 | 1.000 |
| Biomedical eng. | 0.963 | 0.828 | 0.716 | 0.563 | 0.743 | 0.779 | 0.794 |
| Mechanics | 0.943 | 0.829 | 0.829 | 0.559 | 0.706 | 0.647 | 0.941 |
| Materials eng. | 0.987 | 0.920 | 0.760 | 0.595 | 0.811 | 0.784 | 0.932 |
| IT | 0.904 | 0.735 | 0.783 | 0.598 | 0.690 | 0.724 | 0.782 |
| Environmental eng. | 0.957 | 0.739 | 0.855 | 0.652 | 0.797 | 0.754 | 0.928 |
| Energy eng. | 0.953 | 0.802 | 0.791 | 0.628 | 0.826 | 0.767 | 0.872 |
| Electrical eng. | 0.877 | 0.816 | 0.825 | 0.609 | 0.722 | 0.704 | 0.800 |
| Programming | 1.000 | 0.913 | 0.826 | 0.667 | 0.611 | 0.556 | 0.722 |
| Civil eng. | 1.000 | 0.769 | 0.923 | 0.750 | 0.750 | 0.750 | 1.000 |
| Chemical eng. | 0.714 | 0.571 | 0.571 | 0.429 | 0.714 | 0.714 | 0.571 |
| AI | 0.931 | 0.984 | 0.817 | 0.474 | 0.420 | 0.355 | 0.330 |
| Agricultural eng. | 1.000 | 0.867 | 0.800 | 0.705 | 0.864 | 0.795 | 0.932 |
| Aerospace eng. | 1.000 | 0.833 | 1.000 | 1.000 | 0.833 | 0.833 | 1.000 |
| **Science** | | | | | | | |
| Statistics | 0.879 | 0.803 | 0.803 | 0.452 | 0.563 | 0.600 | 0.622 |
| Physics | 0.892 | 0.800 | 0.842 | 0.549 | 0.689 | 0.705 | 0.713 |
| Mathematics | 0.918 | 0.956 | 0.872 | 0.756 | 0.717 | 0.587 | 0.711 |
| Life science | 0.965 | 0.798 | 0.781 | 0.565 | 0.809 | 0.678 | 0.904 |
| Geology | 0.990 | 0.816 | 0.776 | 0.688 | 0.792 | 0.656 | 0.885 |
| Earth science | 0.979 | 0.798 | 0.840 | 0.692 | 0.788 | 0.779 | 0.942 |
| Chemistry | 0.863 | 0.814 | 0.762 | 0.510 | 0.650 | 0.697 | 0.720 |
| Biology | 0.959 | 0.730 | 0.818 | 0.533 | 0.767 | 0.769 | 0.835 |
| Atmospheric science | 0.990 | 0.753 | 0.753 | 0.739 | 0.783 | 0.641 | 0.935 |
| Astronomy | 0.965 | 0.843 | 0.843 | 0.704 | 0.835 | 0.809 | 0.852 |
| **HASS** | | | | | | | |
| Welfare | 0.896 | 0.722 | 0.729 | 0.576 | 0.654 | 0.737 | 0.797 |
| Trade | 0.944 | 0.807 | 0.800 | 0.494 | 0.811 | 0.767 | 0.856 |
| Cognitive studies | 0.620 | 0.524 | 0.481 | 0.500 | 0.580 | 0.662 | 0.629 |
| Religion | 0.912 | 0.877 | 0.895 | 0.724 | 0.914 | 0.860 | 0.948 |
| Politics | 0.909 | 0.759 | 0.693 | 0.635 | 0.767 | 0.767 | 0.872 |
| Philosophy | 0.875 | 0.664 | 0.632 | 0.455 | 0.711 | 0.623 | 0.651 |
| Media | 0.857 | 0.864 | 0.759 | 0.667 | 0.889 | 0.778 | 0.722 |
| Literature | 0.950 | 0.850 | 0.850 | 0.684 | 0.950 | 0.750 | 0.950 |
| Law | 0.750 | 0.596 | 0.610 | 0.294 | 0.540 | 0.518 | 0.679 |
| Language | 0.736 | 0.548 | 0.518 | 0.420 | 0.526 | 0.519 | 0.504 |
| History | 0.911 | 0.864 | 0.578 | 0.463 | 0.786 | 0.857 | 0.881 |
| Geography | 0.911 | 0.804 | 0.804 | 0.628 | 0.773 | 0.886 | 0.818 |
| Education | 0.957 | 0.793 | 0.793 | 0.580 | 0.795 | 0.652 | 0.848 |
| Economics | 0.893 | 0.809 | 0.695 | 0.574 | 0.597 | 0.713 | 0.752 |
| Administration | 0.899 | 0.797 | 0.732 | 0.551 | 0.819 | 0.819 | 0.841 |
| **Social Intelligence** | | | | | | | |
| Value | 0.699 | 0.890 | 0.788 | 0.653 | 0.599 | 0.857 | 0.619 |
| Norms | 0.816 | 0.658 | 0.605 | 0.516 | 0.613 | 0.581 | 0.710 |
| Commonsense | 0.837 | 0.765 | 0.749 | 0.871 | 0.877 | 0.856 | 0.837 |
| Bias | 0.000 | 1.000 | 0.333 | 0.349 | 0.333 | 0.324 | 0.288 |
| **Culture** | | | | | | | |
| Work life | 0.778 | 0.667 | 0.704 | 0.600 | 0.720 | 0.700 | 0.720 |
| Tradition | 0.833 | 0.881 | 0.950 | 0.618 | 0.806 | 0.800 | 0.784 |
| Housing | 1.000 | 1.000 | 0.750 | 1.000 | 1.000 | 0.750 | 0.750 |
| Food | 0.534 | 0.479 | 0.479 | 0.360 | 0.553 | 0.675 | 0.456 |
| Family | 0.913 | 0.739 | 0.609 | 0.591 | 0.659 | 0.705 | 0.818 |
| Daily life | 0.600 | 0.521 | 0.475 | 0.355 | 0.590 | 0.676 | 0.532 |
| Clothing | 1.000 | 1.000 | 1.000 | 1.000 | 1.000 | 1.000 | 1.000 |
| Holiday | 1.000 | 1.000 | 1.000 | 1.000 | 1.000 | 1.000 | 1.000 |
| **Arts % Sports** | | | | | | | |
| Sports | 0.781 | 0.578 | 0.453 | 0.714 | 0.929 | 0.786 | 0.857 |
| Sculpture | 1.000 | 1.000 | 1.000 | 0.500 | 1.000 | 0.500 | 1.000 |
| Photography | 1.000 | 0.600 | 0.800 | 0.400 | 0.400 | 0.800 | 0.800 |
| Performing | 0.846 | 0.846 | 0.769 | 0.673 | 0.654 | 0.808 | 0.846 |
| Painting | 1.000 | 0.600 | 0.900 | 0.600 | 0.900 | 0.700 | 1.000 |
| Music | 1.000 | 1.000 | 0.800 | 0.900 | 0.900 | 0.900 | 0.800 |
| Festivals | 0.500 | 1.000 | 1.000 | 1.000 | 0.500 | 1.000 | 0.500 |
| Fashion | 1.000 | 0.800 | 1.000 | 0.800 | 0.800 | 0.600 | 0.600 |
| Architecture | 1.000 | 0.857 | 0.714 | 0.429 | 1.000 | 0.571 | 1.000 |

Table 18: Results of all models across fine-grained categories (Korean)

| Subject | gpt-4.1 | claude-3.7-sonnet | gemini-2.0 | gemma-3-27b | DeepSeek-R1-32B | Llama-3.3-70B | Mistral-24B |
|---|---|---|---|---|---|---|---|
| **Tech** | | | | | | | |
| Urban eng. | 0.552 | 0.634 | 0.559 | 0.504 | 0.507 | 0.543 | 0.468 |
| Nuclear eng. | 0.676 | 0.647 | 0.618 | 0.676 | 0.559 | 0.588 | 0.588 |
| Marine eng. | 0.688 | 0.826 | 0.625 | 0.569 | 0.521 | 0.611 | 0.569 |
| Biomedical eng. | 0.838 | 0.805 | 0.409 | 0.727 | 0.507 | 0.767 | 0.713 |
| Mechanics | 0.661 | 0.709 | 0.563 | 0.537 | 0.495 | 0.487 | 0.420 |
| Materials eng. | 0.720 | 0.820 | 0.560 | 0.608 | 0.510 | 0.619 | 0.608 |
| IT | 0.854 | 0.877 | 0.667 | 0.727 | 0.756 | 0.803 | 0.742 |
| Environmental eng. | 0.591 | 0.649 | 0.480 | 0.456 | 0.427 | 0.462 | 0.368 |
| Energy eng. | 0.587 | 0.674 | 0.551 | 0.507 | 0.457 | 0.457 | 0.399 |
| Electrical eng. | 0.688 | 0.778 | 0.646 | 0.549 | 0.535 | 0.549 | 0.500 |
| Programming | 0.667 | 0.722 | 0.667 | 0.667 | 0.667 | 0.667 | 0.833 |
| Civil eng. | 0.517 | 0.669 | 0.530 | 0.503 | 0.391 | 0.497 | 0.430 |
| Chemical eng. | 0.711 | 0.809 | 0.641 | 0.596 | 0.539 | 0.574 | 0.560 |
| AI | 0.861 | 0.829 | 0.676 | 0.694 | 0.618 | 0.657 | 0.703 |
| Agricultural eng. | 0.605 | 0.605 | 0.539 | 0.464 | 0.386 | 0.506 | 0.428 |
| Aerospace eng. | 0.757 | 0.786 | 0.579 | 0.621 | 0.564 | 0.629 | 0.579 |
| **Science** | | | | | | | |
| Statistics | 0.813 | 0.813 | 0.571 | 0.571 | 0.582 | 0.549 | 0.615 |
| Physics | 0.826 | 0.870 | 0.644 | 0.626 | 0.595 | 0.603 | 0.542 |
| Mathematics | 0.842 | 0.889 | 0.848 | 0.385 | 0.487 | 0.359 | 0.359 |
| Life science | 0.783 | 0.783 | 0.635 | 0.635 | 0.609 | 0.739 | 0.635 |
| Geology | 0.755 | 0.765 | 0.627 | 0.608 | 0.422 | 0.618 | 0.510 |
| Earth science | 0.701 | 0.769 | 0.627 | 0.604 | 0.552 | 0.575 | 0.575 |
| Chemistry | 0.760 | 0.829 | 0.643 | 0.574 | 0.612 | 0.643 | 0.512 |
| Biology | 0.852 | 0.875 | 0.586 | 0.766 | 0.664 | 0.742 | 0.711 |
| Atmospheric science | 0.719 | 0.688 | 0.625 | 0.531 | 0.531 | 0.656 | 0.563 |
| Astronomy | 1.000 | 1.000 | 1.000 | 0.900 | 1.000 | 1.000 | 0.800 |
| **HASS** | | | | | | | |
| Welfare | 0.783 | 0.745 | 0.516 | 0.755 | 0.742 | 0.724 | 0.705 |
| Trade | 0.856 | 0.767 | 0.658 | 0.752 | 0.752 | 0.766 | 0.731 |
| Religion | 0.846 | 0.860 | 0.714 | 0.805 | 0.706 | 0.812 | 0.856 |
| Psychology | 1.000 | 1.000 | 1.000 | 1.000 | 0.000 | 1.000 | 0.000 |
| Politics | 0.806 | 0.858 | 0.714 | 0.717 | 0.634 | 0.667 | 0.703 |
| Philosophy | 0.843 | 0.897 | 0.715 | 0.791 | 0.718 | 0.757 | 0.757 |
| Media | 0.942 | 0.928 | 0.897 | 0.877 | 0.755 | 0.876 | 0.877 |
| Literature | 0.836 | 0.914 | 0.760 | 0.700 | 0.739 | 0.798 | 0.800 |
| Law | 0.604 | 0.555 | 0.463 | 0.510 | 0.416 | 0.544 | 0.530 |
| Language | 0.807 | 0.906 | 0.763 | 0.648 | 0.685 | 0.750 | 0.705 |
| History | 0.775 | 0.794 | 0.691 | 0.622 | 0.526 | 0.603 | 0.570 |
| Geography | 0.711 | 0.778 | 0.698 | 0.594 | 0.522 | 0.631 | 0.597 |
| Education | 0.732 | 0.816 | 0.586 | 0.701 | 0.603 | 0.755 | 0.660 |
| Economics | 0.814 | 0.820 | 0.606 | 0.704 | 0.701 | 0.692 | 0.656 |
| Administration | 0.731 | 0.766 | 0.598 | 0.691 | 0.635 | 0.711 | 0.675 |
| **Social Intelligence** | | | | | | | |
| Value | 0.848 | 0.879 | 0.697 | 0.818 | 0.818 | 0.788 | 0.758 |
| Norms | 0.884 | 0.881 | 0.881 | 0.881 | 0.810 | 0.721 | 0.762 |
| Commonsense | 0.835 | 0.873 | 0.822 | 0.718 | 0.757 | 0.748 | 0.767 |
| Bias | 0.993 | 0.966 | 0.951 | 1.000 | 1.000 | 0.846 | 1.000 |
| **Culture** | | | | | | | |
| Work life | 0.926 | 0.926 | 0.826 | 0.921 | 0.768 | 0.921 | 0.921 |
| Tradition | 0.962 | 0.960 | 0.858 | 0.917 | 0.819 | 0.900 | 0.911 |
| Leisure | 1.000 | 1.000 | 1.000 | 0.500 | 0.500 | 1.000 | 0.500 |
| Housing | 0.824 | 0.824 | 0.647 | 0.735 | 0.676 | 0.676 | 0.676 |
| Food | 0.850 | 0.923 | 0.769 | 0.744 | 0.684 | 0.789 | 0.821 |
| Family | 0.826 | 0.792 | 0.696 | 0.652 | 0.818 | 0.864 | 0.800 |
| Daily life | 0.837 | 0.837 | 0.823 | 0.751 | 0.682 | 0.738 | 0.764 |
| Clothing | 0.793 | 0.793 | 0.690 | 0.621 | 0.655 | 0.759 | 0.655 |
| Holiday | 0.643 | 0.602 | 0.602 | 0.620 | 0.616 | 0.674 | 0.654 |
| **Arts & Sports** | | | | | | | |
| Sports | 0.960 | 0.960 | 0.818 | 0.960 | 0.917 | 0.913 | 0.864 |
| Sculpture | 0.923 | 0.833 | 0.833 | 1.000 | 0.727 | 0.917 | 0.833 |
| Photography | 0.800 | 0.855 | 0.655 | 0.768 | 0.600 | 0.667 | 0.655 |
| Performing | 0.950 | 0.950 | 0.911 | 0.930 | 0.752 | 0.884 | 0.918 |
| Painting | 0.931 | 0.932 | 0.833 | 0.896 | 0.794 | 0.837 | 0.918 |
| Music | 0.912 | 0.971 | 0.758 | 0.909 | 0.667 | 0.879 | 0.909 |
| Festivals | 0.941 | 1.000 | 1.000 | 0.941 | 0.882 | 0.813 | 0.941 |
| Fashion | 0.626 | 0.626 | 0.524 | 0.565 | 0.490 | 0.571 | 0.456 |
| Architecture | 0.745 | 0.778 | 0.641 | 0.711 | 0.658 | 0.664 | 0.618 |

Tables 19 and 20 details the accuracy of 14 open LLMs across four different sampling strategies in English and Korean, respectively.

Table 21 details the accuracy of 14 open LLMs evaluated by the customized BENCHHUB in five different scenarios.

Table 19: Evaluation results of 14 open LLMs in English across four different sampling strategies

| Model | Random | Stratified | Chatbot Arena | MixEval |
|---|---|---|---|---|
| Qwen2.5-72B-Instruct | 0.874 | 0.962 | 0.888 | 0.870 |
| Qwen3-1.7B | 0.702 | 0.733 | 0.696 | 0.677 |
| Qwen3-14B | 0.743 | 0.778 | 0.733 | 0.729 |
| Qwen3-32B | 0.696 | 0.707 | 0.690 | 0.686 |
| Qwen3-4B | 0.704 | 0.713 | 0.712 | 0.689 |
| Qwen3-8B | 0.732 | 0.749 | 0.737 | 0.723 |
| DeepSeek-R1-Distill-Qwen-14B | 0.729 | 0.763 | 0.726 | 0.707 |
| DeepSeek-R1-Distill-Qwen-32B | 0.746 | 0.799 | 0.755 | 0.743 |
| gemma-3-1b-it | 0.688 | 0.694 | 0.680 | 0.661 |
| gemma-3-27b-it | 0.810 | 0.852 | 0.816 | 0.798 |
| gemma-3-4b-it | 0.717 | 0.748 | 0.721 | 0.704 |
| Llama-3.1-8B-instruct | 0.734 | 0.779 | 0.747 | 0.730 |
| Llama-3.3-70B-Instruct | 0.784 | 0.833 | 0.788 | 0.779 |
| Mistral-Small-24B-Instruct-2501 | 0.817 | 0.845 | 0.811 | 0.789 |

Table 20: Evaluation results of 14 open LLMs in Korean across four different sampling strategies

| Model | Random | Stratified | Chatbot Arena | MixEval |
|---|---|---|---|---|
| Qwen2.5-72B-Instruct | 0.360 | 0.376 | 0.363 | 0.371 |
| Qwen3-1.7B | 0.624 | 0.647 | 0.646 | 0.630 |
| Qwen3-14B | 0.697 | 0.708 | 0.723 | 0.692 |
| Qwen3-32B | 0.597 | 0.613 | 0.547 | 0.606 |
| Qwen3-4B | 0.671 | 0.674 | 0.683 | 0.666 |
| Qwen3-8B | 0.605 | 0.562 | 0.615 | 0.518 |
| DeepSeek-R1-Distill-Qwen-14B | 0.507 | 0.613 | 0.617 | 0.609 |
| DeepSeek-R1-Distill-Qwen-32B | 0.531 | 0.533 | 0.510 | 0.541 |
| gemma-3-1b-it | 0.661 | 0.666 | 0.665 | 0.649 |
| gemma-3-27b-it | 0.453 | 0.474 | 0.469 | 0.468 |
| gemma-3-4b-it | 0.613 | 0.635 | 0.638 | 0.625 |
| Llama-3.1-8B-instruct | 0.612 | 0.623 | 0.618 | 0.623 |
| Llama-3.3-70B-Instruct | 0.370 | 0.444 | 0.383 | 0.406 |
| Mistral-Small-24B-Instruct-2501 | 0.466 | 0.492 | 0.478 | 0.486 |

Table 21: Evaluation results of 14 open LLMs using customized BENCHHUB across five use cases

| Model | (a) | (b) | (c) | (d) | (e) |
|---|---|---|---|---|---|
| Qwen2.5-72B-Instruct | 0.604 | 0.657 | 0.658 | 0.595 | 0.670 |
| Qwen3-1.7B | 0.711 | 0.477 | 0.703 | 0.383 | 0.624 |
| Qwen3-4B | 0.667 | 0.420 | 0.556 | 0.300 | 0.599 |
| Qwen3-8B | 0.629 | 0.568 | 0.718 | 0.430 | 0.665 |
| Qwen3-14B | 0.642 | 0.429 | 0.531 | 0.316 | 0.499 |
| Qwen3-32B | 0.798 | 0.523 | 0.663 | 0.529 | 0.648 |
| DeepSeek-R1-Distill-Qwen-14B | 0.657 | 0.554 | 0.653 | 0.479 | 0.647 |
| DeepSeek-R1-Distill-Qwen-32B | 0.626 | 0.609 | 0.654 | 0.488 | 0.660 |
| Llama-3.1-8B-Instruct | 0.650 | 0.581 | 0.602 | 0.393 | 0.627 |
| Llama-3.3-70B-Instruct | 0.651 | 0.612 | 0.637 | 0.562 | 0.659 |
| Mistral-Small-24B-Instruct-2501 | 0.619 | 0.632 | 0.661 | 0.523 | 0.660 |
| gemma-3-1b-it | 0.762 | 0.465 | 0.704 | 0.364 | 0.551 |
| gemma-3-4b-it | 0.632 | 0.529 | 0.641 | 0.391 | 0.632 |
| gemma-3-27b-it | 0.611 | 0.614 | 0.651 | 0.582 | 0.664 |

