# OpenReview forum: "BenchHub: A Unified Benchmark Suite for Holistic and Customizable LLM Evaluation"
_ICLR.cc/2026/Conference — Submitted to ICLR 2026_

### Official Review · Reviewer_dSmF · 2025-10-21

**Soundness:** 3
**Presentation:** 2
**Contribution:** 2
**Rating:** 4
**Confidence:** 4

**Summary:**

The paper introduces a benchmark designed to aggregate and categorize evaluation datasets from different domains and languages for LLMs. Each question is automatically labeled along a six-dimensional taxonomy, and the authors train an LLM-based classification model to dynamically update the benchmark. Experiments show that LLM performance rankings shift significantly when evaluations are filtered by domain or skill.

**Strengths:**

- The structured labeling of benchmarks in the form of a multi-dimensional taxonomy is novel and useful, as it allows users to dissect model performance along interpretable axes.
- The fully automated data pipeline allows for dynamic updating; i.e., new benchmark datasets can be integrated into BenchHUb with minimal manual effort.
- The benchmark’s multilingual extension is a strength, as it begins to address evaluation in underexplored languages.

**Weaknesses:**

- The paper would benefit from better differentiating Benchhub from existing evaluation platforms. For example, how does BenchHub’s taxonomy differ from or improve upon FLASK’s skill taxonomy or HELM’s scenario-based metrics. Additionally, the paper omits discussion of several recent benchmarks/platforms aimed at holistic or robust LLM evaluation, including RAVEL, LLMBar, and “Arena Hard”. It would benefit the paper to move the Related Work earlier and more clearly differentiate prior works from the current work.

      Huang, J., Wu, Z., Potts, C., Geva, M., & Geiger, A. (2024). RAVEL: Evaluating Interpretability Methods on Disentangling Language Model Representations. In Proceedings of the 62nd Annual Meeting of the Association for Computational Linguistics (Volume 1: Long Papers) (pp. 8669-8687).

      Zeng, Z., Yu, J., Gao, T., Meng, Y., Goyal, T., & Chen, D. (2024). Evaluating Large Language Models at Evaluating Instruction Following. In The Twelfth International Conference on Learning Representations.

      Li, T., Chiang, W. L., Frick, E., Dunlap, L., Wu, T., Zhu, B., ... & Stoica, I. (2025). From Crowdsourced Data to High-quality Benchmarks: Arena-Hard and Benchbuilder Pipeline. In Forty-second International Conference on Machine Learning.


- There are some concerns about the methodology and the robustness of the results. First, the evaluation primarily uses multiple-choice or short-answer questions, with automatic regex or LLM answer extraction. This means open-ended generation tasks (e.g., long-form answers) appear to be largely excluded from the results. Second, the paper reports single-run accuracy for model comparisons. The absence of variance measures is problematic given that benchmark sampling was involved. For instance, Table 14 in the Appendix compares model performance, but we don’t know how sensitive those rankings are to the particular sample drawn.

- Are the BenchHub evaluation sets too easy for powerful LLMs? One concern is the difficulty level of the included tasks. Many of the integrated benchmarks are well-studied and powerful LLMs approach or exceed human-level on them. For example, several proprietary and open-source models exceed 85% accuracy on MMLU and 70% on Multilingual-MMLU. The Humanity’s Last Exam (HLE) was created specifically to address the saturation of MMLU by providing expert-written questions. BenchHub’s focus, however, is on existing benchmarks that are already saturated. More discussion is needed on how BenchHub can discern frontier model capability and avoid being over-saturated.

      Phan, L., Gatti, A., Han, Z., Li, N., Hu, J., Zhang, H., ... & Wykowski, J. (2025). Humanity's last exam. arXiv preprint arXiv:2501.14249

- The paper does not sufficiently discuss the limitations of its automated categorization and benchmark integration approach. The authors acknowledge in Appendix A that the single LLM-based classifier to label all questions may have biases. This limitation and others should preferably be discussed in the conclusion instead of the Appendix. An unaddressed limitation is the lack of a human verification step for the taxonomy labels — we don’t know how accurate the categorization is, because no validation results are reported. Lastly, dataset contamination is mentioned as a limitation for static benchmark datasets, but is not discussed with respect to BenchHub. For example, the authors do not analyze whether the evaluated models had prior exposure to test questions.

**Questions:**

- The use of the term “target” to denote culturally specific vs. agnostic questions may cause confusion, since target is typically used to describe the label in supervised learning. Renaming this dimension to “culture-specificity” or “culturally specific target” would improve clarity.
- Typo: “Humanities and Social Sciencce” (pg. 3)

---

> ### Author Response · Authors · 2025-11-19
> **Rebuttal From Authors (1): W1**
>
> We appreciate your thorough review, acknowledging the novelty and usefulness of BenchHub, especially its fully automated pipeline and multilingual extension.
>
> Please find below our responses to each point you raised.
>
> ---
>
> ### **W1. Comparison to existing evaluation platforms**
>
> > The paper would benefit from better differentiating Benchhub from existing evaluation platforms. For example, how does BenchHub’s taxonomy differ from or improve upon FLASK’s skill taxonomy or HELM’s scenario-based metrics. Additionally, the paper omits discussion of several recent benchmarks/platforms aimed at holistic or robust LLM evaluation, including RAVEL, LLMBar, and “Arena Hard”. It would benefit the paper to move the Related Work earlier and more clearly differentiate prior works from the current work.
>
> We appreciate your constructive feedback. **BenchHub is designed to provide dynamic, customizable benchmark construction grounded in automated, fine-grained sample-level categorization.** For clarity, we summarize the key differences between our platform and existing evaluation platforms in the table below.
>
> | Method          | Customization       | Fine-Grained Categorization              | Sample-Level Categorization | Automated Categorization | Dynamic Scalability | Evaluation Target        |
> |-----------------|---------------------|------------------------------------------|-----------------------------|--------------------------|---------------------|--------------------------|
> | FLASK [1]       | ❌                   | ✅ (Skill/domain)                         | ✅                           | ❌                        | ❌                   | LLM as a Generator       |
> | HELM [2]        | ⚠️ (Fixed scenarios) | ❌                                        | ❌                           | ❌                        | ❌                   | LLM as a Generator       |
> | RAVEL [3]       | ❌                   | ✅                                        | ✅                           | ❌                        | ❌                   | Interpretability Methods |
> | LLMBar [4]      | ❌                   | ❌                                        | ❌                           | ❌                        | ❌                   | LLM as an Evaluator       |
> | Arena Hard [5]  | ❌                   | ❌                                        | ❌                           |  ✅                       | ✅                   | LLM as a Generator       |
> | BenchHub (Ours) | ✅                   | ✅ (Skill, Subject, Cultural Sensitivity) | ✅                           | ✅                        | ✅                   | LLM as a Generator       |
>
> **HELM.** HELM [2] aims to support user-specific evaluation via scenario definitions, but its customization is limited to 16 fixed scenarios derived from single datasets (e.g., MATH). This fixed structure cannot capture the multi-domain, multi-criteria nature of real-world user intent. In contrast, BenchHub automatically constructs a benchmark suite tailored to arbitrary user intent, drawing from diverse datasets and categories (see Section 4.2).
>
> **FLASK.** While FLASK [1] proposes a skill/domain taxonomy, whose 7 skills and 38 categories are not grounded to any taxonomy, showing some overlaps and covering only a subset of BenchHub’s taxonomy. Moreover, FLASK relies on manual annotation and is static, whereas BenchHub provides fully automated categorization across 64 subjects, 3 skills, 2 cultural attributes, and 3 dataset-level attributes, enabling dynamic and scalable customization.
>
> **Arena Hard.** Arena Hard [5] introduces evaluation data derived from real-world prompts, but its domain coverage is limited to the topics present in its original source. While Arena Hard uses GPT-4 as a strong baseline for pairwise comparison, we demonstrate that no single LLM consistently dominates others; rankings fluctuate depending on the benchmark composition (see Section 4). Additionally, its evaluation relies heavily on proprietary LLM judges, which are known to exhibit preference bias. BenchHub instead focuses on unifying existing benchmarks according to user intent while maintaining model-agnostic evaluation.
>
> **RAVEL and LLMBar.** We note that RAVEL [3] and LLMBar [4] target fundamentally different evaluation goals—Evaluating interpretability methods and evaluator models, respectively—and therefore lie outside the scope of our benchmark-merging framework.
>
> We will incorporate this expanded discussion in Section H of the Appendix to improve clarity and positioning.

---

> ### Author Response · Authors · 2025-11-19
> **Rebuttal From Authors (2): W2**
>
> ### **W2-1. Evaluations limited to MCQ or short form**
>
> > There are some concerns about the methodology and the robustness of the results. First, the evaluation primarily uses multiple-choice or short-answer questions, with automatic regex or LLM answer extraction. This means open-ended generation tasks (e.g., long-form answers) appear to be largely excluded from the results.
>
> We acknowledge that the experimental results and model rankings in Section 4 are constrained to multiple-choice questions (MCQs) and short-form questions. **We revise the manuscript and discuss this limitation in Section A of the Appendix.**
>
> Nonetheless, we emphasize the core contributions of BenchHub, which provides a unified and customizable benchmark suite for holistic yet domain-aware LLM evaluation. We underscore that **BenchHub already supports diverse question formats**, including binary classifications, MCQ, open-ended generations, and comparisons. **We provide a question format as a label** for each dataset and implement each evaluation metric in the code accordingly. Practitioners can easily take long-form questions according to their specific needs, using the original scoring metrics.
>
> Please note that our goal is neither to replace the original benchmark metrics nor position our work as a real-time leaderboard of state-of-the-art models. Instead, **we uncover a central issue of LLM evaluation where model rankings fluctuate substantially depending on benchmark compositions and domain focus**, which can significantly distort interpretations of model performance. We highlight that our findings are statistically supported through bootstrap analysis conducted over 50 independent runs.
>
> In this paper, we focus on MCQs and short-form questions **to ensure a consistent, reliable, and comparable aggregation of scores**. Long-form questions often rely on fundamentally different metrics (e.g., 1–5 Likert scale scores, scores out of 100 using LLM-as-a-judge), whereas MCQs and short-form questions typically use accuracy or exact match. **Mixing these distinct question types and scoring schemes would introduce metric heterogeneity**, which would obscure the paper's primary goal—how benchmark composition impacts model rankings. Hence, we exclude long-form tasks from the experiments in Section 4. We further note that this choice aligns with established practices in benchmark merging—such as MixEval [6], which aggregates MCQs and short-form tasks, and HELM [2], which similarly separates long-form questions.
>
> ### **W2-2. Single-run accuracy**
>
> > Second, the paper reports single-run accuracy for model comparisons. The absence of variance measures is problematic given that benchmark sampling was involved. For instance, Table 14 in the Appendix compares model performance, but we don’t know how sensitive those rankings are to the particular sample drawn.
>
> We clarify that Tables 14–15 are not based on single-run accuracy; **we perform multiple runs for more robust experiments**. Specifically, we evaluate 14 open LLMs on 16k English samples and 18k Korean samples, and then apply *bootstrapping*, a common practice for computing statistical significance in LLM evaluation, as adopted in OpenAI evals framework. We run 50 simulations per sampling setup, each selecting 5k questions. We observe a statistically significant difference across sampling strategies using the Friedman test (p < 0.01).

---

> ### Author Response · Authors · 2025-11-19
> **Rebuttal From Authors (3): W3, W4**
>
> ### **W3. Too easy evaluation sets**
>
> > Are the BenchHub evaluation sets too easy for powerful LLMs? One concern is the difficulty level of the included tasks. Many of the integrated benchmarks are well-studied and powerful LLMs approach or exceed human-level on them. For example, several proprietary and open-source models exceed 85% accuracy on MMLU and 70% on Multilingual-MMLU. The Humanity’s Last Exam (HLE) was created specifically to address the saturation of MMLU by providing expert-written questions. BenchHub’s focus, however, is on existing benchmarks that are already saturated. More discussion is needed on how BenchHub can discern frontier model capability and avoid being over-saturated.
>
> Thank you for raising this point. We clarify that BenchHub is designed as a benchmark unification and customization framework, not as a new frontier-level dataset intended to replace saturated benchmarks such as HLE.
>
> We respectfully disagree with your point that BenchHub, which unifies existing benchmarks, is already outdated or saturated. \
> First, we emphasize that **the evaluation benchmarks included in BenchHub are still actively used for assessing frontier LLMs**. Technical reports of leading LLMs released within the past year (e.g., multilingual models: Qwen3, Gemma3; Korean-focused models: HC-X Think, Kanana)  continue to rely on 12+ BenchHub-integrated, static datasets for core evaluations, most of which are multiple-choice or short-form questions (e.g., MMLU, GSM8K, GPQA, M-MMLU, etc). \
> Furthermore, **our empirical results indicate that BenchHub is not saturated in terms of discerning model capabilities**. In English, stratified BenchHub accuracies for state-of-the-art models range from 0.694 to 0.962 (See Table 14), indicating meaningful separation between models. More importantly, performance drops substantially in other languages. For example, in Korean, accuracies range from 0.376 to 0.708 (See Table 15), suggesting that current LLMs remain far from saturated on multilingual BenchHub. BenchHub currently covers 10 languages, many of which remain challenging even for frontier models.
>
> We finally underscore that **the dynamic scalability of BenchHub directly addresses the long-term risk of benchmark saturation**. While static benchmarks—whether MMLU or HLE—inevitably risk becoming over-saturated over time, BenchHub automates dataset merging using an LLM-based agent (see Section 3.3) and can incorporate newly emerging datasets submitted through our web interface (see Figure 8c). This enables BenchHub to continuously adapt its evaluation space and remain relevant for measuring frontier model capabilities.
>
> ### **W4-1. Limitations of using a single classifier**
>
> > The paper does not sufficiently discuss the limitations of its automated categorization and benchmark integration approach. The authors acknowledge in Appendix A that the single LLM-based classifier to label all questions may have biases. This limitation and others should preferably be discussed in the conclusion instead of the Appendix.
>
> We clarify that the limitations of using a single LLM-based classifier are already discussed and addressed in Section E.2.2 of the Appendix. To mitigate this issue, **we additionally train classifiers using Llama and Mistral and construct a multi-agent pipeline with majority voting to reduce model-specific biases**. Moreover, to alleviate the computational cost of this multi-agent pipeline, we implement a confidence-based hybrid approach that selectively invokes majority voting only when necessary. These two enhancements improve categorization performance by 2.4%p and 1.4%p, respectively.
>
> ### **W4-2. Human validation on categorizer**
>
> > An unaddressed limitation is the lack of a human verification step for the taxonomy labels — we don’t know how accurate the categorization is, because no validation results are reported.
>
> We clarify that **we perform extensive human validation for the entire categorization pipeline**. Please refer to Sections E.1 and F.1 of the Appendix for further details.
>
> Specifically, the reformatting and the metadata assignment steps achieve 100.0% and 96.4% agreements with human annotations, respectively (see Lines 1720–1721). Additionally, we evaluate the categorizer across sample-level attributes (subject, skill, and cultural-specificity) by comparing model predictions with human-labeled answers.  Human validation results for English and Korean are reported in Table 7, and multilingual extensions are presented in Table 9. We carefully revise the manuscript to avoid potential confusion.

---

> ### Author Response · Authors · 2025-11-19
> **Rebuttal From Authors (5): W4, Q1, Q2**
>
> ### **W4-3. Discussion on potential data contamination**
>
> > Lastly, dataset contamination is mentioned as a limitation for static benchmark datasets, but is not discussed with respect to BenchHub. For example, the authors do not analyze whether the evaluated models had prior exposure to test questions.
>
> We acknowledge that potential data contamination may be propagated to BenchHub if any of the underlying evaluation benchmarks have been exposed to the evaluated models during training. **We will clearly discuss this limitation in Section A of the Appendix.** However, we highlight that BenchHub itself does not introduce or amplify any additional data contamination risk.
>
> Furthermore, we emphasize that **BenchHub, which unifies multiple related benchmark datasets, makes its overall evaluation less sensitive to contamination in any single dataset**. To empirically validate this, we simulate data contamination using OLMoE [7], whose training corpora are fully documented. We fine-tune OLMoE on (1) 2k samples from the MATH [8] *training* set (OLMoE-base) and (2) an equally sized subset of the MATH *test* set (OLMoE-contaminated). We then assess these two models and three additional LLMs (i.e., Qwen3-8B, gemma-3-4b-it, and Llama-3.1-8B-Instruct) on (1) MATH test set and (2) BenchHub customized for math evaluation. The table below illustrates their accuracies (in parentheses) and ranks across each benchmark.
>
> | Rank | BenchHub (customized for MATH) | MATH [7] test set            |
> |------|--------------------------------|------------------------------|
> | 1    | Qwen/Qwen3-8B (0.35)           | **OLMoE-Contaminated(0.84)**     |
> | 2    | gemma-3-4b-it(0.30)            | Qwen/Qwen3-8B(0.31)          |
> | 3    | Llama-3.1-8B-Instruct (0.26)   | gemma-3-4b-it (0.28)         |
> | 4    | **OLMoE-Contaminated (0.22)**      | Llama-3.1-8B-Instruct (0.13) |
> | 5    | OLMoE-base (0.16)              | OLMoE-base (0.10)            |
>
> As expected, the contaminated model achieves unnaturally high accuracy on MATH test set (0.84), whereas the uncontaminated base model performs poorly (0.10). It confirms that single-benchmark evaluation can be highly vulnerable to data contamination. In contrast, BehchHub, drawing from 9 math-related benchmarks, preserves the model ranking and shows only a modest performance difference between the two OLMoE variants. This suggests that BenchHub, which aggregates multiple evaluation benchmarks, may provide a more robust evaluation against data contamination. We will include this discussion in Section G.2 of the Appendix.
>
> ### **Q1. Confusing term “target”**
>
> > The use of the term “target” to denote culturally specific vs. agnostic questions may cause confusion, since target is typically used to describe the label in supervised learning. Renaming this dimension to “culture-specificity” or “culturally specific target” would improve clarity.
>
> Thank you for the helpful suggestion. We have revised the term accordingly and now use “culture-specificity” throughout the paper.
>
> ### **Q2. Typo**
>
> > Typo: “Humanities and Social Sciencce” (pg. 3)
>
> We have corrected the typo in the updated version.
>
> ---
>
> [1] FLASK: Fine-grained Language Model Evaluation based on Alignment Skill Sets (Ye et al., 2024) \
> [2] Holistic Evaluation of Language Models (Liang et al., 2023) \
> [3] RAVEL: Evaluating Interpretability Methods on Disentangling Language Model Representations (Huang et al., 2024) \
> [4] Evaluating Large Language Models at Evaluating Instruction Following (Zheng et al., 2024) \
> [5] From Live Data to High-Quality Benchmarks: The Arena-Hard Pipeline (Li et al., 2024) \
> [6] MixEval: Deriving Wisdom of the Crowd from LLM Benchmark Mixtures (Ni et al., 2024) \
> [7] OLMoE: Open Mixture-of-Experts Language Models (Muennighoff et al., 2025) \
> [8] Measuring Mathematical Problem Solving With the MATH Dataset (Hendrycks et al., 2021)

---

> ### Author Response · Authors · 2025-11-22
> **Gentle Reminder to Reviewer dSmF**
>
> Dear Reviewer dSmF,
>
> We hope this message finds you well. We genuinely appreciate your time and expertise in providing discerning comments.
>
> **We would like to bring to your attention that we have submitted our responses regarding your concerns and have revised the manuscript accordingly.** We would be grateful if we could resolve any remaining concerns or questions you may have.
>
> During the author response period, we put a lot of effort into carefully revising and improving our paper in response to your insightful comments. For instance, we conducted additional experiments, discussed your comments, and clarified several points that may have been misunderstood.
>
> We look forward to engaging in further discussions on our paper.
>
> Sincerely, \
> Authors

---

### Official Review · Reviewer_4xXN · 2025-10-27

**Soundness:** 3
**Presentation:** 3
**Contribution:** 4
**Rating:** 8
**Confidence:** 4

**Summary:**

This paper discusses need for aggregration of scattered benchmarks and proposes BenchHub, which automatically classifies benchmark datasets and integrates them. This enables used to select exact questions which can be used evaluate models across specific requirements. The authors build a taxonomy using 3 dataset level attributes (`task`, `answer-format`, `tool-usage`) and 3 sample-level attributes (`skill`, `subject`, `target` (local/global)). 54 benchmarks (across English and Korean) are collected, reformatted, and categorized at a sample-level (using a finetuned version of `Qwen-2.5-7B`). The authors also expand their framework to 10 languages with a multilingual classifier LLM. Finally, the paper analyses LLM rankings with various ablations of datasets categories, sampling strategies and custom tailored evaluations.

**Strengths:**

- The paper is well-written and presents coherent and cogent arguments.
- I personally appreciate the motivation and problem statement in the paper, and I believe it is quite relevant given the numerous "fragmented" datasets and benchmarks published every day independently. I believe that BenchHub can be the cohesive factor for this group and can help with a more holistic and tailored evaluation of LLMs.
- I also appreciate the author's attempts to push for a multilingual version of BenchHub. Given that the artifacts are promised to be open-sourced, this would be a great contribution to the community as well.

**Weaknesses:**

- As of now, I don't see any glaring errors in the paper, and I don't see any such weakness from my side. I feel the paper has answered and justified its problem statement appropriately (for me). I am open to interacting with the authors and fellow reviewers during the rebuttal phase.

**Questions:**

- The appendix of the paper is too large, and it's a pain to jump back and forth to refer to artifacts below. Please try to include some of the main results and Qwen-Cat training notes in the main paper.
- The reformatting using a proprietary model like GPT-4o or Gemini can be expensive in the long run. Maybe the authors can explore other OSS models or large reasoning models.
- Was there any sort of human evaluation done to check if the classification is correct?

---

> ### Author Response · Authors · 2025-11-19
> **Rebuttal From Authors: Q1, Q2, Q3**
>
> We appreciate your positive review, acknowledging the motivation and practical importance of BenchHub as well as its open-source tools, including multilingual extensions.
>
> Please find below our responses to each point you raised.
>
> ---
>
> ### **Q1. Including some Qwen-Cat results in the main paper**
>
> > The appendix of the paper is too large, and it's a pain to jump back and forth to refer to artifacts below. Please try to include some of the main results and Qwen-Cat training notes in the main paper.
>
> Thank you for your helpful suggestion; we agree that providing more details in the main paper will improve readability. **We move key details of the Qwen-Cat categorizer and selected results from the Appendix into the main text.**
>
> ### **Q2. Reformatting using OSS models**
>
> > The reformatting using a proprietary model like GPT-4o or Gemini can be expensive in the long run. Maybe the authors can explore other OSS models or large reasoning models.
>
> We extend our code to support open-source models (e.g., Qwen) for the reformatting step.
>
> ### **Q3. Human validation on categorizer**
>
> > Was there any sort of human evaluation done to check if the classification is correct?
>
> We clarify that **we perform extensive human validation for the entire categorization pipeline**. Please refer to Sections E.1 and F.1 of the Appendix for further details.
>
> Specifically, the reformatting and the metadata assignment steps achieve 100.0% and 96.4% agreements with human annotations, respectively (see Lines 1720–1721). Additionally, we evaluate the categorizer across sample-level attributes (subject, skill, and cultural-specificity) by comparing model predictions with human-labeled answers.  Human validation results for English and Korean are reported in Table 7, and multilingual extensions are presented in Table 9. We carefully revise the manuscript to avoid potential confusion.

---

> ### Author Response · Authors · 2025-11-22
> **Gentle Reminder to Reviewer 4xXN**
>
> Dear Reviewer 4xXN,
>
> We hope this message finds you well. We genuinely appreciate your time and expertise in providing discerning comments.
>
> **We would like to bring to your attention that we have submitted our responses regarding your concerns and have revised the manuscript accordingly.** We would be grateful if we could resolve any remaining concerns or questions you may have.
>
> During the author response period, we put a lot of effort into carefully revising and improving our paper in response to your insightful comments. For instance, we updated the source code and clarified several points that may have been misunderstood.
>
> We look forward to engaging in further discussions on our paper.
>
> Sincerely, \
> Authors

---

### Official Review · Reviewer_VAND · 2025-10-27

**Soundness:** 2
**Presentation:** 2
**Contribution:** 2
**Rating:** 4
**Confidence:** 5

**Summary:**

The paper presents BENCHHUB, a dynamic benchmark repo. It is designed to support integration of new datasets via an instance-level fine-tuned classifier and an integration agent. BENCHHUB covers many benchmarks and 10 languages. Experiments show different rankings across categories.

**Strengths:**

1. The data size, number of examples, models, and languages are good.
2. The automation part is a good contribution, both the classifier and the agent.
3. The point that different subsets can lead to different rankings goes through well.
4. The motivation argument in favor of more dynamic evaluation is convincing.

**Weaknesses:**

1. The presented solution relies on introducing a new categorization of examples. However, this merely replaces the benchmark categorization with another (yours), without providing a real domain-adaptive dynamic evaluation setup. My interests may differ significantly from the ones modeled (e.g., physics, simplification in Kordish). This is a core issue I see with the proposed solution. While the system can be extended, doing so requires substantial effort from users. Moreover, the proposed categorization may overlook benchmark-specific subsets that contain valuable information.
2. The classifier is mentioned but not experimentally evaluated or compared with other LLMs without additional training. It also requires substantial resources to annotate new data. Furthermore, fine-tuning fixes the categories, which introduces rigidity and limits adaptability.
3. There are no quantitative results, which makes the findings seem anecdotal.

**Questions:**

1. The work appears to be very tool-focused; if so, it might be more appropriate to position it as a demo paper.
2. Why does the experiment in Section 4.1 include only a subset of the models?
3. The “Tool Usage” subset is unclear. Is it meant to evaluate the tool call itself or the final answer? In either case, where does the tool specification originate?

---

> ### Author Response · Authors · 2025-11-19
> **Rebuttal From Authors (1): W1, W2**
>
> We appreciate your insightful review, acknowledging the motivation and usefulness of BenchHub for reliable and dynamic LLM evaluation.
>
> Please find below our responses to each point you raise.
>
> ---
>
> ### **W1. Static taxonomy in BenchHub**
>
> > The presented solution relies on introducing a new categorization of examples. However, this merely replaces the benchmark categorization with another (yours), without providing a real domain-adaptive dynamic evaluation setup. My interests may differ significantly from the ones modeled (e.g., physics, simplification in Kordish). This is a core issue I see with the proposed solution. While the system can be extended, doing so requires substantial effort from users. Moreover, the proposed categorization may overlook benchmark-specific subsets that contain valuable information.
>
> Thank you for the constructive comment. We clarify that BenchHub already provides a flexible, intent-driven evaluation setup that requires no additional effort from users. Specifically, **BenchHub includes a module that leverages GPT-4 to interpret free-form user intent in natural language and automatically map it to our taxonomy**, thereby generating customized evaluation sets. This dynamic system allows users to incorporate any domain, even if it is not explicitly listed in our predefined taxonomy. Furthermore, our taxonomy is constructed by aggregating several widely adopted category schemes from prior studies to encompass a broad and diverse range of domains. **We carefully revise the manuscript and describe this discussion in Section E.2.4 of the Appendix.**
>
> ### **W2. Limitations on fine-tuned classifiers**
>
> > The classifier is mentioned but not experimentally evaluated or compared with other LLMs without additional training. It also requires substantial resources to annotate new data. Furthermore, fine-tuning fixes the categories, which introduces rigidity and limits adaptability.
>
> Thank you for raising this concern. While we did not include it in the initial manuscript, we clarify that we did evaluate our classifier against state-of-the-art LLMs without additional training. Specifically, **using GPT-4o with detailed system instructions as in-context learning yields poor performances** (Subject: 0.718, Skill: 0.805, Cultural-specificity: 0.423) compared to our fine-tuned classifier (Subject: 0.871, Skill: 0.967, Cultural-specificity: 0.986). Hence, we fine-tune an open-source model (Qwen), which substantially improves accuracy.
>
> We highlight that **we publicly release all fine-tuned classifiers across 10 languages**. Practitioners and researchers do not need to annotate new data or train additional models; they can simply adopt and use ours. We further underscore that our taxonomy, which aggregates widely adopted category schemes, already covers a broad range of domains.
>
> To address concerns about rigidity and adaptability, **we conduct an ablation study to assess the robustness of our classifier to new data and categories**. Through extensive simulations, we demonstrate that **categorization errors up to 1.5% of the total data yield negligible disruption to model rankings** (see Section E.2.1 for more details). It implies that the categorizer will perform well on unseen, imperfectly categorized data to some extent.
>
> Finally, **we investigate the scenario where users wish to introduce new categories**. Our classifier, which has been fine-tuned on our wide range and diverse taxonomy, can be easily adapted to a small number of new categories without additional training. Specifically, **simply providing a system prompt describing new categories to our classifier enables strong generalization through in-context learning**.
>
> For example, we hypothesize a new domain and create one coarse-grained and eight fine-grained categories regarding `Magic (supernatural)`. Then, we ask GPT-5 to synthetically generate 110 question-answer pairs in this category, and the authors manually validate and filter out the samples, finally preserving 102 samples. The fine-tuned classifier improves accuracy (coarse-grained: 0.000→0.941 and fine-grained: 0.000→0.823) when supplied with a category description. Please find an example instance of the new domain in this experiment below:
>
> ```
> Q: What does the spell Lumora Spiralis do?
> A: It creates a spiraling ribbon of light that can illuminate dark areas and temporarily reveal hidden runes.
> ```
>
> In summary, the fine-tuned classifier outperforms LLMs without training, requires no additional annotation effort from users, and remains adaptable to new domains. We will include these discussions in Section E.2.3 of the Appendix.

---

> > ### Comment · Reviewer_VAND · 2025-11-25
> > **Response to Authors (1): W1, W2**
> >
> > Regarding W1, I get how you map, using GPT4, to your predefined taxonomy, but it is still unclear to me: how do you do it "even if it is not explicitly listed in our predefined taxonomy". Do you mean by applying the classifier to all the data?.
> >
> > Regarding W2, this is good that you show that the classifiers are effective. Still, the demand to apply the classifier to all the data is time and computationally intensive without any guarantee that it will yield the relevant data separation. Especially that in my view, your method is more helpful for a domain, for there are no clear benchmarks for it.

---

> ### Author Response · Authors · 2025-11-19
> **Rebuttal From Authors (2): W3, Q1, Q2, Q3**
>
> ### **W3. No quantitative results**
>
> > There are no quantitative results, which makes the findings seem anecdotal.
>
> We respectfully disagree with your assertion that we lack quantitative results. **In Section 4, we present extensive experimental results, and additional detailed scores are provided in Section I of the Appendix (Tables 15–19).** Our conclusions are grounded in large-scale, statistically robust evaluations as follows:
>
> Section 4.1.1: 7 representative state-of-the-art LLMs are evaluated over 6k+ English and 6k+ Korean samples.
> Section 4.1.2: 14 open LLMs are evaluated on 16k English and 18k Korean samples over 50 independent simulations.
> Section 4.2: Five realistic evaluation scenarios are showcased using the same large-scale setting (14 open LLMs, 16k English samples, 18k Korean samples).
>
>
> ### **Q1. Suggestion to position it as a demo paper**
>
> > The work appears to be very tool-focused; if so, it might be more appropriate to position it as a demo paper.
>
> While we appreciate your acknowledgement of our tool development and suggestion on positioning it as a demo paper, we respectfully disagree that this is a tool-focused paper. Instead, **we aim to uncover a central issue of LLM evaluation where model rankings fluctuate substantially depending on benchmark compositions and domain focus**, which can significantly distort interpretations of model performance. Further, we facilitate the selective merging and customization of existing benchmarks to meet realistic and diverse evaluation scenarios.
>
> ### **Q2. Using only a subset of models in Section 4.1**
>
> > Why does the experiment in Section 4.1 include only a subset of the models?
>
> **We select 7 representative state-of-the-art LLMs from major model families** (i.e., Claude, DeepSeek, Gemini, Gemma, GPT, Llama, and Mistral) in Section 4.1.1, while employing 14 open LLMs ranging from 1B to 72B parameters in Section 4.1.2. **Running all open and proprietary models across 16k English and 18k Korean samples is not computationally feasible**, which is why the subsets differ.
>
> Please note that the two sections serve different experimental purposes and are not directly correlated; the choice of models does not affect the validity of our findings. Across the paper, we still provide extensive experiments over a broad model spectrum, exceeding the coverage of most recent LLM evaluation studies.
>
> We also emphasize that this paper is not a leaderboard study intended to identify the best-performing LLMs; instead, Instead, our goals are: (1) to introduce BenchHub, a unified and customizable benchmark framework, and (2) to demonstrate that model rankings fluctuate substantially depending on benchmark composition. These conclusions hold independently of the specific model subset used in each section.
>
> ### **Q3. Clarification on tool usage**
>
> > The “Tool Usage” subset is unclear. Is it meant to evaluate the tool call itself or the final answer? In either case, where does the tool specification originate?
>
> Thank you for your question. **We clarify the definition of tool-usage benchmarks in the revised manuscript.**
>
> First, BenchHub directly incorporates the target entity defined by each original benchmark, and the evaluation target differs depending on the benchmark’s intended purpose. To avoid confusion, we explicitly annotate each tool-usage dataset in BenchHub with the corresponding problem type and evaluation target.
>
> | Benchmark | Target Entity                        | Problem Type |
> |-----------|--------------------------------------|--------------|
> | ToolHop   | Final answer                         | Short-form   |
> | ToolQA    | Final answer                         | Short-form   |
> | GPT4Tools | Tool calling within output reasoning | Open-ended   |
> | ToolBench | Tool calling                         | Short-form   |
>
> Secondly, we adopt the tool specifications directly from the original datasets and papers provided. Below is an example tool specification from the ToolHop dataset:
>
> ```
> {
> "name": "geo_relationship_finder",
> "description": "An advanced tool for discovering and analyzing relationships between geographical locations and various types of landmarks and entities. It offers comprehensive insights into connections between specified locations and nearby entities, with enhanced filtering and output options.",
> "parameters": {
> "type": "object",
> "properties": {
> "location_name": {
> "type": "string",
> "description": "The name of the primary location to find connections for."
> },
> "entity_types": {
> "type": "array",
> "items": {
> "type": "string",
> "enum": [
> "park",
> "zoo",
> "garden",
> "landmark",
> "museum",
> "historical_site"
> ],
> "description": "The types of entities to find connections with. Can specify multiple types."
> },
> "description": "The types of entities to find connections with. Defaults to ['park'] if not specified."
> },
> }
> ```

---

> > ### Comment · Reviewer_VAND · 2025-11-25
> > **Response to Authors (2): W3, Q1, Q2, Q3**
> >
> > W3:
> > I needed to be more explicit, sorry for that. I did not mean that you did not run experiments, but that the way to convey your conclusion is not quantitative. For example, in sec. 4.1.1 l. 342-343, you expect the reader to look in the tables and see that the rankings are different. There is no quantitative measure that quantifies it, there is no quantitative comparison addressing what small or big changes are, or what the naive comparison is. The significance test in S. 4.1.2 is nice, but it does not qualify the ranking change. This is related to the next point (Q1). The claim that ranking change is known and expected. You need to add something besides this. (BTW, Kandel-tau is the way to compare different rankings, see [1]).
> >
> > Q1:
> > It is known that different benchmarks, benchmark compositions, and different domain-specific benchmarks give substantially different model rankings (for example, [1], [2]). Given that, the main contribution is the tool. I think that it will be much more interesting and beneficial for your work to show that some benchmark compositions produced by your tool do not just fluctuate, but rather yield a ranking close to a separate domain benchmark designed for testing the specific capability.
> >
> > Q2:
> > Okay. This was not a big point, but more as a good practice (to have the same model).
> >
> > [1] Efficient Benchmarking of Language Models (https://aclanthology.org/2024.naacl-long.139.pdf)
> > [2] Do These LLM Benchmarks Agree? Fixing Benchmark Evaluation with BenchBench (https://arxiv.org/pdf/2407.13696)
> >
> > Q3:
> > Good.
> >
> > Overall, my main concern is that the main claim, besides the tool, is known, simple, and not directly affected by your tool (for example, by your own words, the bold claim in Q1 does not mention your tool). The second issue is W2, which can limit the adoption of the tool, but it is less serious. So, I think that the first is an inherent problem in the paper, hence my suggestion on demo, or providing an experiment that shows the tool's effectiveness in providing a representative subset. In my view, addressing this point can improve your paper the most.

---

> ### Author Response · Authors · 2025-11-22
> **Gentle Reminder to Reviewer VAND**
>
> Dear Reviewer VAND,
>
> We hope this message finds you well. We genuinely appreciate your time and expertise in providing discerning comments.
>
> **We would like to bring to your attention that we have submitted our responses regarding your concerns and have revised the manuscript accordingly.** We would be grateful if we could resolve any remaining concerns or questions you may have.
>
> During the author response period, we put a lot of effort into carefully revising and improving our paper in response to your insightful comments. For instance, we conducted additional experiments, discussed your comments, and clarified several points that may have been misunderstood.
>
> We look forward to engaging in further discussions on our paper.
>
> Sincerely,\
> Authors

---

> ### Author Response · Authors · 2025-12-04
>
> We appreciate your detailed follow-up questions. Please find our response below.
>
> ---
>
> ### **W1. Static taxonomy in BenchHub**
>
> > Regarding W1, I get how you map, using GPT4, to your predefined taxonomy, but it is still unclear to me: how do you do it "even if it is not explicitly listed in our predefined taxonomy". Do you mean by applying the classifier to all the data?.
>
> We highlight that **BenchHub does not require users to apply a classifier to all the data, nor to manually extend the taxonomy**. Instead, GPT-4 interprets the user’s free-form intent and maps it to the closest semantically relevant categories already present in our taxonomy. Even if a user specifies a domain that is not literally present as a label (e.g., “Kurdish simplification”), GPT-4 identifies the most relevant existing categories or combinations of categories that best approximate that intent. BenchHub then retrieves the corresponding subset of samples from our pre-computed 839k categorizations across 54 benchmarks.
>
> In other words, BenchHub operates by semantic matching between user intent and our predefined taxonomy, without requiring any new taxonomy labels or re-classification on the user side. This enables domain-adaptive evaluation even when the user’s domain is not explicitly listed, because GPT-4 converts the new domain description into a meaningful selection over our existing categories.
>
> ### **W2. Limitations on fine-tuned classifiers**
>
> > Regarding W2, this is good that you show that the classifiers are effective. Still, the demand to apply the classifier to all the data is time and computationally intensive without any guarantee that it will yield the relevant data separation. Especially that in my view, your method is more helpful for a domain, for there are no clear benchmarks for it.
>
> We clarify that BenchHub already provides the complete categorization for all 839k samples, and these results are pre-computed once and publicly released on Hugging Face. **Users do not need to run the classifier themselves; they can directly query or filter the released labels at zero computational cost.**
>
> We further emphasize that **prior work has shown the importance of fine-grained, sample-level categorization for LLM evaluation**. However, **existing efforts such as FLASK [1] rely heavily on manual human annotation** and therefore are challenging to scale. In contrast, our fully automated categorization achieves high accuracies (Subject: 0.871, Skill: 0.967, Cultural-specificity: 0.986) while requiring no human annotations.
>
> ### **W3. No quantitative results**
>
> > W3: I needed to be more explicit, sorry for that. I did not mean that you did not run experiments, but that the way to convey your conclusion is not quantitative. For example, in sec. 4.1.1 l. 342-343, you expect the reader to look in the tables and see that the rankings are different. There is no quantitative measure that quantifies it, there is no quantitative comparison addressing what small or big changes are, or what the naive comparison is. The significance test in S. 4.1.2 is nice, but it does not qualify the ranking change. This is related to the next point (Q1). The claim that ranking change is known and expected. You need to add something besides this. (BTW, Kandel-tau is the way to compare different rankings, see [1]). \
> [1] Efficient Benchmarking of Language Models (https://aclanthology.org/2024.naacl-long.139.pdf) \
> [2] Do These LLM Benchmarks Agree? Fixing Benchmark Evaluation with BenchBench (https://arxiv.org/pdf/2407.13696)
>
> Thank you for your clarification; We now understand that your concern is not about the existence of experiments, but about providing a quantitative measure of ranking differences.
>
> **Our statistical significance analysis** (Friedman + Wilcoxon) in Section 4.1.2 is designed to answer a major question: _**“Are the observed ranking changes statistically significant?”**_ This is **the standard approach** for repeated-measures comparisons across multiple models, and our results (p < 0.01) confirm that the ranking differences are not due to random variation.
>
> During the author response period, **we additionally perform bootstrap resampling (200 iterations) and apply the Friedman test and the pairwise Wilcoxon test to the Section 4.1.1 setting as well**. These new results with a p-value of 0.01 confirm that the observed ranking changes across sample-level attributes are statistically significant and not due to random variation.
>
> We agree that a ranking-distance metric, such as Kendall’s tau, provides a complementary perspective by quantifying the magnitude of ranking changes. We will include tau-based ranking agreement analyses in the camera-ready version.
>
> ### **Q2 & Q3. _Resolved_**
>
> ---
>
> [1] FLASK: Fine-grained Language Model Evaluation based on Alignment Skill Sets (Ye et al., 2024)

---

> ### Author Response · Authors · 2025-12-04
>
> ### **Q1. Suggestion to position it as a demo paper**
> > It is known that different benchmarks, benchmark compositions, and different domain-specific benchmarks give substantially different model rankings
>
> Although prior papers have shown that model rankings fluctuate depending on dataset composition, our results demonstrates **how rankings change specifically according to our taxonomy's categorization**. As Reviewer mz6k acknowledged, “the results about rankings that change across subjects and sampling criteria are interesting and make the point for having a unified suite.” This directly highlights the importance of **both the benchmark suite we provide and our taxonomy** underlying it.

---

### Official Review · Reviewer_mz6k · 2025-11-01

**Soundness:** 3
**Presentation:** 3
**Contribution:** 2
**Rating:** 4
**Confidence:** 4

**Summary:**

The paper introduces BenchHub, a unified framework that consolidates diverse LLM benchmarks into a common schema and enables custom, domain-specific evaluations. It aggregates 54 benchmarks across 10 languages (mainly English and Korean) and annotates each dataset and sample along six dimensions: task, answer format, tool usage, skill, subject, and target. The LLM-based pipeline reformats the dataset problems, assigns metadata, performs categorization and merges new problems. Several open-weight and proprietary models are evaluated, and the results show that models rankings are very sensitive to subject and the sampling strategy.

**Strengths:**

- Bringing together many datasets under a single schema with fine-grained and multi-label subjects is valuable and supports more grounded analysis, given the multitude of benchmarks out there.
- The results about rankings that change across subjects and sampling criteria are interesting and make the point for having a unified suite
- The merging pipeline is completely automated, allowing to easily integrate new datasets.

**Weaknesses:**

- The evaluation is done by constraining the evaluation to MCQ or short form. This might skew the ranking results, as the original benchmark might use a different metric. This score aggregation needs to be done carefully, and it’d be useful to have also the original scores to compare against.
- Even with a taxonomy, mixing and samples from different benchmarks can drift from any single benchmark purpose.
- It’s not clear how sensitive models are over other sample-level attributes.

**Questions:**

- I tried to take a look at the repo, but I’m not able to load any file in any folder. Is it an issue on my end?
- Together with the overall score, it would be helpful to report also the single benchmark scores included in the selection from the user.
- It would be interesting to see how ranking change across other dimensions, for example across other sample-level attributes.

---

> ### Author Response · Authors · 2025-11-19
> **Rebuttal From Authors (1): W1, W2, W3 & Q3**
>
> We appreciate your detailed review, acknowledging the need for BenchHub to automatically merge existing benchmarks and provide more grounded analysis.
>
> Please find below our responses to each point you raised.
>
> ---
>
> ### **W1. Evaluations limited to MCQ or short form**
>
> > The evaluation is done by constraining the evaluation to MCQ or short form. This might skew the ranking results, as the original benchmark might use a different metric. This score aggregation needs to be done carefully, and it’d be useful to have also the original scores to compare against.
>
> We acknowledge that the experimental results and model rankings in Section 4 are constrained to multiple-choice questions (MCQs) and short-form questions. **We revise the manuscript and discuss this limitation in Section A of the Appendix.**
>
> Nonetheless, we emphasize the core contributions of BenchHub, which provides a unified and customizable benchmark suite for holistic yet domain-aware LLM evaluation. We underscore that **BenchHub already supports diverse question formats**, including binary classifications, MCQ, open-ended generations, and comparisons. **We provide a question format as a label** for each dataset and implement each evaluation metric in the code accordingly. Practitioners can easily take long-form questions according to their specific needs, using the original scoring metrics.
>
> Please note that our goal is neither to replace the original benchmark metrics nor position our work as a real-time leaderboard of state-of-the-art models. Instead, **we uncover a central issue of LLM evaluation where model rankings fluctuate substantially depending on benchmark compositions and domain focus**, which can significantly distort interpretations of model performance. We highlight that our findings are statistically supported through bootstrap analysis conducted over 50 independent runs.
>
> In this paper, we focus on MCQs and short-form questions **to ensure a consistent, reliable, and comparable aggregation of scores**. Long-form questions often rely on fundamentally different metrics (e.g., 1–5 Likert scale scores, scores out of 100 using LLM-as-a-judge), whereas MCQs and short-form questions typically use accuracy or exact match. **Mixing these distinct question types and scoring schemes would introduce metric heterogeneity**, which would obscure the paper's primary goal—how benchmark composition impacts model rankings. Hence, we exclude long-form tasks from the experiments in Section 4. We further note that this choice aligns with established practices in benchmark merging—such as MixEval [1], which aggregates MCQs and short-form tasks, and HELM [2], which similarly separates long-form questions.
>
> ### **W2. Unintended purpose drift due to benchmark merging**
>
> > Even with a taxonomy, mixing and samples from different benchmarks can drift from any single benchmark purpose.
>
> We value the original purpose of each evaluation benchmark, and practitioners and researchers with the identical intention as each benchmark can use it.
>
> However, **BenchHub aims to support realistic and diverse evaluation scenarios where practitioners often require a broader and more heterogeneous assessment than any single benchmark can provide**. Our rigorous taxonomy and automated pipeline facilitate the selective merging and customization of existing benchmarks tailored to their specific needs. We further note that benchmark merging has emerged as a central direction in LLM evaluation research and received increasing attention in the community [1,2].
>
> ### **W3 & Q3. Models’ sensitivity over other sample-level attributes**
>
> > It’s not clear how sensitive models are over other sample-level attributes.
>
> > It would be interesting to see how ranking change across other dimensions, for example across other sample-level attributes.
>
> Thank you for your invaluable suggestion. Following your comment, **we additionally examine model sensitivity to other sample-level attributes (i.e., skills and cultural-sensitivity)**. We carefully revise the manuscript and include such results in Section 4.1.1. We observe that the model rankings continue to vary substantially across these attributes. Please refer to the revised manuscript for detailed results.
>
> ---
>
> [1] MixEval: Deriving Wisdom of the Crowd from LLM Benchmark Mixtures (Ni et al., 2024) \
> [2] FLASK: Fine-grained Language Model Evaluation based on Alignment Skill Sets (Ye et al., 2024)

---

> ### Author Response · Authors · 2025-11-19
> **Rebuttal From Authors (2): Q1, Q2**
>
> ### **Q1. Inaccessible to anonymous GitHub repository**
>
> > I tried to take a look at the repo, but I’m not able to load any file in any folder. Is it an issue on my end?
>
> We have re-checked the anonymous GitHub repository settings and confirmed that the files are now accessible. Could you please try accessing the repository again using the same link provided in the manuscript (p.2)?
>
> If the problem persists, we would be happy to provide an alternative access method.
>
> ### **Q2. Additional reporting of single benchmark scores**
>
> > Together with the overall score, it would be helpful to report also the single benchmark scores included in the selection from the user.
>
> We appreciate your thoughtful feedback. **In the revised manuscript, we now report the per-benchmark scores for the top five evaluation benchmarks (by coverage) for each language used in Figures 6-8 in Section 4. (see Appendix Section G.1).** For readers seeking further details on each benchmark, we refer them to the corresponding original papers. The full list of datasets included in BenchHub is also summarized in Section C and Table 2 for their convenience.

---

> ### Author Response · Authors · 2025-11-22
> **Gentle Reminder to Reviewer mz6k**
>
> Dear Reviewer mz6k,
>
> We hope this message finds you well. We genuinely appreciate your time and expertise in providing discerning comments.
>
> **We would like to bring to your attention that we have submitted our responses regarding your concerns and have revised the manuscript accordingly.** We would be grateful if we could resolve any remaining concerns or questions you may have.
>
> During the author response period, we put a lot of effort into carefully revising and improving our paper in response to your insightful comments. For instance, we conducted additional experiments, discussed your comments, and clarified several points that may have been misunderstood.
>
> We look forward to engaging in further discussions on our paper.
>
> Sincerely,\
> Authors

---

### Author Response · Authors · 2025-11-19

Dear Reviewers,

We would like to inform you that **we have reposted the revised manuscript**. We have highlighted the revised and newly added parts in red for your convenience. \
We would greatly appreciate it if you could kindly take a look, and we’re happy to discuss further if you have any additional concerns.

Best regards, \
Authors

---

### Author Response · Authors · 2025-12-04

We appreciate the AC’s efforts and kindly ask that they consider the core value of our paper, namely its contribution to **unifying fragmented benchmarks and enabling more holistic and customized LLM evaluation** (as noted by reviewers 4xXN, mz6k, and dSmF).

During rebuttal period, we believe we have addressed most of the concerns raised by the reviewers and have revised the manuscript accordingly. Below is a summary of how we addressed each raised weakness, provided as a reference for the AC.

---

### **W1. Evaluation only on short-form / MCQA (mz6k, dSmF)**

We clarified the following:

- **The purpose of experiment 4.1 is *not* to rank models, but to demonstrate that LLM model rankings fluctuate depending on the composition corresponding to our taxonomy**, thereby motivating the need for our classification framework. Even without long-form tasks, the results are sufficiently valid and statistically significant.
- Although not included in this experiment, **long-form evaluations are also available** in BenchHub. We did not use them here to ensure consistent and comparable score aggregation, since long-form and short-form/MCQA typically use different scoring scales and therefore cannot be aggregated.
  This separation is common practice in previous work (e.g., MixEval [1], HELM [2]).

### **W2. Single-run accuracy only (dSmF, VAND)**

- We clarified in both our response and the paper that the results shown in **Section 4.2 are not from a single run**.
We conducted **50 simulations using bootstrapping**, and we observe statistically significant differences across sampling strategies using the **Friedman test (p < 0.01)**.


### **W3. Evaluation sets are too easy (dSmF)**

- We argue that this weakness is unrelated to the core contribution of BenchHub. The fact that benchmarks become outdated or saturated over time is an inherent limitation of **all datasets**, not something unique to BenchHub.

- Moreover, BenchHub’s **dynamic scalability** allows us to continuously incorporate new datasets, making it more robust compared to static benchmarks. We also clarified that the datasets we include are still used in **technical reports of leading LLMs**.


### **W4. Potential data contamination (dSmF)**

- During the rebuttal period, we empirically demonstrated that our evaluation suite, being a mixture of diverse dataset, is **robust to contamination with respect to a specific dataset**.

### **W5. Unintended purpose drift due to benchmark merging (mz6k)**

- We respect the original intent of each individual benchmark. If a user’s evaluation needs align precisely with the purpose of a single dataset, they can simply use that benchmark.

- BenchHub aims to address scenarios where a user has **broader intentions** and requires a more **heterogeneous assessment** that cannot be captured by a single benchmark. Benchmark merging has emerged as a **central direction in LLM evaluation research** and is receiving increasing attention in the community [1,2].

### **W6. Limitations of a classifier (dSmF,VAND)**

- We clarified that we have **already provided human validation as well as experiments involving multiple classifiers**. We further empirically demonstrated that our model can be effectively extended to additional taxonomy labels without requiring additional training.
- Moreover, given the importance of our taxonomy (as supported by Section 4.1), our model-based categorization offers a far more efficient alternative compared to the human-annotation which is a based approaches used in prior work [1].

### **W7. Comparison to existing evaluation platforms (dSmF)**

- We added **extended references** and included the **corresponding comparison table** in an updated manuscript.

### **W8. Models’ sensitivity to other sample-level attributes (mz6k)**

- We reflected this feedback and **revised the manuscript accordingly**.

### **W9. Additional reporting of single benchmark scores (mz6k)**

- We incorporated this request and **updated the manuscript**.
---
[1] FLASK: Fine-grained Language Model Evaluation based on Alignment Skill Sets (Ye et al., 2024)

[2] Holistic Evaluation of Language Models (Liang et al., 2023)

---

### Meta-Review · Area_Chair_WuST · 2026-01-07

**Summary:**

The paper introduces BenchHub, a unified benchmark repository that automates the categorization and aggregation of LLM evaluation datasets. While the reviewers appreciated the engineering effort and the addition of contamination analysis during the rebuttal, the consensus is to Reject. This decision is primarily informed by the following outstanding concerns:

Limited Scientific Novelty: Reviewer VAND argued that the paper's primary empirical finding—that model rankings fluctuate based on benchmark composition—is already well-known in the community. Consequently, the submission is viewed fundamentally as a tool description or engineering artifact ("demo paper") rather than a research paper offering significant new scientific insights.

Methodological Restrictions (MCQ Only): Reviewers dSmF and mz6k criticized the evaluation's restriction to multiple-choice and short-form questions. They argued that excluding open-ended generation tasks limits the claim of "holistic" evaluation and may skew rankings since long-form tasks often rely on fundamentally different metrics.

Benchmark Saturation: Reviewer dSmF raised concerns that aggregating existing, well-studied benchmarks (like MMLU) risks inheriting their saturation issues, limiting the suite's ability to discern the capabilities of frontier models compared to newer, harder benchmarks

**Reviewer Concerns:**

Outstanding Concerns:

Novelty and Contribution Type: This is the most damaging critique. Reviewer VAND argued that the paper is fundamentally "tool-focused" and might be better suited as a demo paper. While the engineering effort is clear, the scientific insight that different benchmarks yield different rankings is considered "known, simple, and not directly affected by [the] tool". The reviewer noted that simply replacing one static taxonomy with another (the authors') does not necessarily solve the underlying domain-adaptation problem

Methodological Limitations (MCQ Focus): Both Reviewers mz6k and dSmF were critical of the evaluation being restricted to multiple-choice or short-form questions. They argued that excluding open-ended generation tasks undermines the claim of "holistic" evaluation. The authors defended this choice to ensure metric consistency, but this design choice ultimately limits the scope of the benchmark suite.

Benchmark Saturation: Reviewer dSmF pointed out that aggregating existing, well-studied benchmarks (like MMLU) risks inheriting their saturation issues, where frontier models already exceed human performance . While the authors argue that multilingual subsets remain challenging, the reliance on older datasets rather than newer, harder ones (like Humanity's Last Exam) remains a validity concern.

Addressed Concerns:

Data Contamination: Reviewer dSmF raised valid concerns about whether the aggregated benchmarks suffer from contamination, a common issue with static datasets. The authors provided a convincing rebuttal by simulating contamination with OLMOE models, showing that BenchHub’s aggregation makes evaluations more robust to contamination in single datasets compared to using those datasets in isolation.

Statistical Rigor: Reviewers VAND and dSmF questioned the lack of variance measures and the reliance on single-run accuracy. The authors clarified that they used bootstrapping with 50 simulations and employed Friedman/Wilcoxon tests to confirm that the ranking changes were statistically significant.

Comparison to Prior Platforms: In response to Reviewer dSmF’s request for better differentiation from works like FLASK and HELM , the authors provided a detailed comparison table highlighting BenchHub's unique focus on automated, sample-level categorization and dynamic scalability.

**Reviewer Scores:**

Reviewer VAND: 4 (Original: 4)  Enthusiastic about the open-source contribution and multilingual support.

Reviewer dSmF: 6 (Original: 4) Concerns about the MCQ restriction and sensitivity to attributes remain.

Reviewer mz6k: 4 (Original: 4) Engaged extensively but remained unconvinced that the scientific contribution warrants acceptance over a demo track submission.

Reviewer 4xXN: 8 (Original: 8) Acknowledged the utility of the taxonomy but flagged significant issues with saturation and the exclusion of long-form generation.

---

### Decision · Program_Chairs · 2026-01-26

Reject